

# Sensitivity of tropical woodland savannas to El Niño droughts
Simone Matias Reis[1,2,3*], Yadvinder Malhi[1], Ben Hur Marimon Junior[2], Beatriz Schwantes
Marimon[2], Huanyuan Zhang-Zheng[1], Renata Freitag[2], Cécile A. J. Girardin[1], Edmar Almeida de
Oliveira[2], Karine da Silva Peixoto[2], Luciana Januário de Souza[2], Ediméia Laura Souza da Silva[2],
Eduarda Bernardes Santos[2], Kamila Parreira da Silva[2], Maélly Dállet Alves Gonçalves[2], Cecilia
A. L. Dahlsjö[1], Oliver Lawrence Phillips[4], Imma Oliveras Menor[1,5]
[1]School of Geography and the Environment, Environmental Change Institute, University of
Oxford, Oxford, UK
[2]Laboratório de Ecologia Vegetal/Programa de Pós-graduação em Ecologia e Conservação,
Universidade do Estado de Mato Grosso, Nova Xavantina, BR
[3]Centro de Ciências Biológicas, Universidade Federal do Acre, Rio Branco, BR
[4]School of Geography, University of Leeds, Leeds, UK
[5]AMAP (Botanique et Modélisation de l'Architecture des Plantes et des Végétations), Université
de Montpellier, CIRAD, CNRS, INRAE, IRD, Montpellier, France
Correspondence to: Simone Matias Reis (simonematiasreis@gmail.com)
**Abstract**
The 2015-2016 El Niño event led to one of the most intense and hottest droughts for many tropical
forests, profoundly impacting forest productivity. However, we know little about how this event
affected the Cerrado, the largest savanna in South America. Here we report 5 years of productivity
of the dominant vegetation types in Cerrado, savanna (*cerrado*) and transitional forest-savanna
(*cerradão*), continuously tracked before, during, and after the El Niño. We carried out intensive
monitoring between 2014 and 2019 of the productivity of key vegetation components (stems,
leaves, roots). Before the El Niño total productivity was ~25% higher in the *cerradão* compared
to the *cerrado*. However, *cerradão* productivity declined strongly by 29% during the El Niño event.
The most impacted component was stem productivity, reducing by 58%. By contrast, *cerrado*
productivity varied little over the years, and while the most affected component was fine roots,
declining by 38% during the event, fine root productivity recovered soon after the El Niño. The
two vegetation types also showed contrasting patterns in the allocation of productivity to canopy,
wood, and fine-root production. Our findings demonstrate that *cerradão* can show low resistance
and resilience to climatic disturbances due to the slow recovery of productivity. This suggests that
the transitional Amazon-Cerrado ecosystems between South America's largest biomes may be
particularly vulnerable to drought enhanced by climate change.
**Keywords:** 2015-2016 El Niño, productivity, productivity allocation, climate events, *cerradão*,
*cerrado*.

## 1 Introduction

The 2015-2016 El Niño event led to one of the most intense droughts of the last century as well
as record maximum temperatures, coming on top of decades of long-term warming (Jiménez-



Muñoz et al., 2016; Liu et al., 2017). The 2015-16 climate anomaly affected most of the tropics
but was especially potent in Amazonia (Gloor et al., 2018). Intense droughts can increase tree
mortality and affect the carbon sequestration capacity of forests as shown by long-term ground-
based monitoring (e.g., Phillips et al., 2009; Feldpausch et al., 2016; Rifai et al., 2018; Bennett et
al., 2023). Satellite-based analyses also reveal the impacts of climate anomalies on carbon
dynamics (Palmer et al., 2018; Fan et al., 2019), providing a synoptic view of ecosystem
productivity. However, we still lack ground-based, tree-level measurements of net primary
productivity (NPP) through extreme tropical climate events, hindering our understanding of key
aspects of the vegetation carbon cycle response, such as recovery following drought events, and
NPP allocation. Measuring these ecosystem responses directly is helped by tracking long-term
forest dynamics in permanent plots but especially requires high-fidelity process-based
measurements sustained over time. These are exceptionally challenging to make and require
long-term dedication to measurements before, during, and after major climate events like the
2015-16 El Nino. We know especially little about how the productivity of savanna ecosystems is
affected by El Niño events, especially in the extensive Amazonia-Cerrado transition in South
America.
The Amazonia-Cerrado transitional region contains a mixture of Amazonia and Cerrado
species, making the species composition of this region unique and diverse (Ratter et al., 1973;
Marimon et al., 2006; Morandi et al., 2016). Despite its ecological importance, this region has
been greatly impacted by deforestation (~41% between 1984 and 2014) so that today only
fragments of native vegetation remain (e.g., Marques et al., 2020). In recent decades, the
remaining vegetation has been affected by increasing temperatures, frequent wildfires, extreme
drought events, and the long-term lengthening of the dry season (e.g., Reis et al., 2018; Silvério
et al., 2019; Nogueira et al., 2019; Matricardi et al., 2020; Araújo et al., 2021a). Deforestation,
together with increases in temperature and reduction in precipitation during El Niño events,
increases wildfire occurrence and carbon emissions, reducing the capacity of the vegetation to
act as a carbon sink (Covey et al., 2021; Gatti et al., 2021). As the Amazonia-Cerrado transition
is the driest, warmest, and most fragmented region in the Amazon basin (e.g., Matricardi et al.,
2020; Marques et al., 2020; Covey et al., 2021; Reis et al., 2022) it is especially vital to understand
better how climate change and extreme climate events impact carbon dynamics.
This transition is composed naturally of a mosaic of vegetation, being the typical cerrado
(referred to as *cerrado* hereafter) and woodland savanna (i.e., *cerradão*) the most common in the
regions (Ratter et al., 1973; Marimon et al., 2006, Oliveras & Malhi, 2016). Despite co-existing in
the same space, *cerrado* and *cerradão* vegetation formations show contrasting characteristics
(Marimon-Junior & Haridasan, 2005; Marimon et al., 2006). The *cerradão* is a transitional forest-
savanna characterized by closed canopy, understory formed by small shrubs and herbs, with few
grasses, and average height of the tree stratum varying from 8 to 15 m, tree cover of 50 to 90%
(Ribeiro & Walter, 2008, Oliveras & Malhi 2016), while *cerrado* is a savanna vegetation type with
a discontinuous canopy, trees, and shrubs with grass understorey, and a low average height of



just 3 to 6 m, with tree cover of 20 to 50% (Marimon-Junior & Haridasan, 2005; Ribeiro & Walter,
81 2008).

In the *cerrado,* most species are deciduous, fully shedding their leaves in the dry season,
while most *cerradão* species are brevi-deciduous. Although the dominant species of both
vegetation types show strong stomatal efficiency (Jancoski et al., 2022), trees in the *cerrado* have
smaller stomata and higher trichome density than individuals occurring in the *cerradão*,
anatomical features that help the leaves minimise their water loss (Araújo et al., 2021b; Araújo et
al., 2023). Furthermore, for species that co-occur in both *cerrado* and *cerradão,* individuals in
*cerrado* lose their leaves earlier than *cerradão* in the dry season. The early loss of leaves in the
*cerrado* means that the photosynthetic apparatus is not harmed during the driest and hottest
period of the year. In the *cerradão*, individuals take longer to lose their leaves, which makes them
more sensitive to changes in temperature increases, both current and projected (Araújo et al.,
2021b). *Cerradão* trees are taller than *cerrado* trees, this characteristic may offer *cerradão* greater
sensitivity to drought, since taller trees have wider xylem vessels (Araújo et al., *under review*).
These characteristics (e.g., larger stomata and greater maximum stomatal pore opening) may
give the *cerradão* greater sensitivity to disturbances generated by climatic anomalies, such as the
2015-2016 El Niño.
Here, by setting up and sustaining intensive, long-term monitoring plots that experience
a similar climate at *cerradão* and *cerrado*, we aimed to quantify and compare the effect of the
2015/2016 El Niño on the carbon cycle (productivity and allocation) of these two vegetation types.
Our guiding questions are: 1) Does productivity and allocation differ between *cerradão* and
*cerrado*? 2) How did the 2015-2016 El Niño affect productivity and allocation in the *cerradão* and
*cerrado*? 3) Did the *cerradão* and *cerrado* regain productivity after the El Niño? 4) What are the
trade-offs in resource allocation between canopy, wood and fine roots during drought in the two
vegetation types?

**2 Materials and Methods**
*2.1 Study sites*
We conducted this study in two long-term plots in *cerradão* (transitional forest-savanna – NXV-
02; Forestplots code) and *cerrado* (*typical cerrado* – savanna – NXV-01) located in the Bacaba
Municipal Park, in Nova Xavantina, Mato Grosso State, Central Brazil. The park covers
approximately 500 ha in the transition zone between the Cerrado (Brazilian savanna) and
Amazonia. Since the two plots are about 300m apart, they experience very similar climates, which
corresponds to the *Aw* (tropical with dry winters) type in Köppen's classification system (Alvares
et al., 2013). As measured by station #83319 of the Brazilian National Institute of Meteorology
(INMET), the mean monthly temperature is 24.8 °C, the total annual precipitation is 1440 mm
(Peixoto et al., 2017), and the average altitude of the park is ~ 250 m (Marimon Junior &
Haridasan, 2005).
The plots were established in 2002 (Marimon Junior & Haridasan, 2005) and have been
re-censused multiple times. Since 2010, these have been part of the PELD (Cerrado-Amazonia





Forest Transition: ecological and socio-environmental bases for Conservation), RAINFOR (Amazonia Forest Inventory Network; ForestPlots.net et al., 2021) and ForestPlots.net collaborations, and since 2014, part of GEM (Global Ecosystems Monitoring network; Malhi et al., 2021). The plots have facilitated multiple studies, such as soil, composition and diversity of species, biomass, and tree dynamics (e.g., Marimon Junior & Haridasan, 2005; Marimon et al., 2014). Partial data on carbon cycling have previously been published for the *cerradão* plot, on litterfall, soil efflux and carbon stocks at fine roots, litter layer, and stem (Peixoto et al., 2017; Peixoto et al., 2018). Here we provide the first comprehensive description of the carbon cycling in both plots as well as an extended time series that provides insight into the aftermath of 2015/2016 El Niño event.

The plots have not been burned since 2008. The *cerradão* plot is a transitional forest-savanna characterized by the overlap of savanna and forest species, a closed canopy, and with dominant species (notably *Hirtella glandulosa* Spreng. and *Tachigali vulgaris* L.G. Silva & H.C. Lima). This type of *cerradão* was recognised by Ratter et al. (1973) as *Hirtella glandulosa cerradão*. Trees and shrubs with grass understorey and open canopy characterize the *cerrado*. Here the two dominant tree species are *Qualea parviflora* Mart. and *Davilla elliptica* A.St.-Hil. (Marimon Junior & Haridasan, 2005; Marimon et al., 2014). The vegetation of the *cerrado* is becoming denser and there are not many grasses present (Morandi et al., 2015), possibly due to fire exclusion.

The soil is similar across the plots – sandy loams of the yellow latosol type, acidic (pH < 5.0) and dystrophic ($Ca^{2+}$ ~ 0.4 $cmol_c$ $kg^{-1}$), with high levels of exchangeable aluminium ($Al^{3+}$ > 1.3 $cmol_c$ $kg^{-1}$) – however, the *cerradão* soil presents higher percentages of clay and potential water holding capacity than the *cerrado* (Marimon Junior & Haridasan, 2005). These differences in soil texture may explain the different vegetation formations in these two closely adjacent sites. The average height of the trees in *cerrado* is 3.7 m, and a basal area of ~14.9 m² $ha^{-1}$. For the *cerradão*, the average tree height is 6.4 m and basal area of ~ 21.4 m² $ha^{-1}$ (Marimon Junior & Haridasan, 2005). The species number was 77 in both and the number of trees similar (*cerrado* = 1890 and *cerradão* = 1884) (Marimon Junior & Haridasan, 2005).

### *2.2 Site climate and the El Niño 2015/2016 event*

We used the climate variables – air temperature, relative air humidity, and precipitation – in time series from a Meteorological Station (World Weather Station 83319), about 800 m from the plots. We calculated the maximum climatological water deficit (MCWD), a climatological measure of tropical forest water stress (see Aragão et al., 2007). To calculate MCWD, we considered a standardized evapotranspiration (ET) value for wet season tropical forests of 100 mm month-1 (Aragão et al., 2007).

The seasonality of the plots is marked by two well-defined seasons – cooler-dry (April to September) and hot-rainy (October to March). We defined the twelve months from May 2015 to April 2016 as the climate of the 2015-2016 El Niño Southern Oscillation event based on Liu et al. (2017). During the El Niño, the plots experienced record mean and mean monthly maximum



annual temperatures (26.0 °C and 35.4 °C) and record low total annual precipitation (790.2 mm),
and in September 2016, record low annual MCWD (-883.7 mm) (Fig. 1; Table S1).

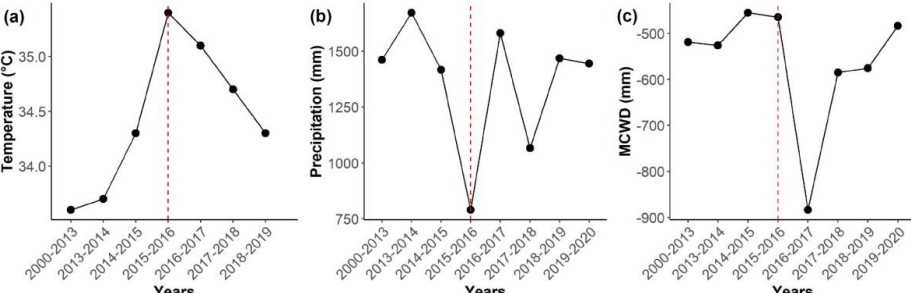


**Fig. 1. Climate variables** between 2000 and 2020 for *cerrado* and *cerradão*. We should (a)
temperature (°C), (b) precipitation (mm/year) and (c) maximum climatological water deficit
(MCWD, mm in a rolling year) with the 1st month of the dry season (May) representing the
beginning of each year's climatic calendar. The temperature indicates the average maximum
monthly temperatures. The dashed red line indicates the El Niño periods. Climatic data are from
meteorological station #83319 of the Brazilian National Institute of Meteorology (INMET). See
Table S1 for data.

***2.3 Field methods***
We followed the GEM protocol manual (Marthews et al., 2014; Malhi et al., 2021) to collect the
data for this study. We measured the main components of NPP: canopy (leaves, twigs,
reproductive parts, and others), wood (stems and branches), and fine roots (see Table 1).

**Table 1**. Field methods for intensive monitoring of NPP components from *cerradão* (NXV-02) and
*cerrado* (NXV-01) plots in the transition zone between Amazonia and Cerrado. See also the
RAINFOR-GEM manual (Marthews et al., 2014). nc= no collected.

| Component | Description | Sampling period | Sampling interval |
|---|---|---|---|
| **Above-ground net primary productivity ($NPP_{AG}$) and biomass** | | | |
| Aboveground coarse wood net primary productivity ($NPP_{stem}$) and Stem biomass | **Forest inventory** or plot census were done in the years listed. The default measurement point was set at 30cm ($DAS_{30cm}$) above soil surface, instead of a typical forest diameter at breast height at 1.3m. All trees ≥ 5 cm $DAS_{30cm}$ were censused, based on which, we calculated mortality and recruitment rate of new trees. **Stem biomass** for each tree was calculated. The sum of all alive trees in each census is termed stem biomass. As we have noticed trees stem shrinking, we calculate stem NPP as the change of alive trees biomass between two censuses, where dead and new recruit trees are excluded but shrinking trees are included. We have also presented 'Stem Diameter growth' which excludes shrinking trees, dead trees and new recruit trees. In 2014, five 10 m x 10 m subplots were established to census small trees (≤ 5 cm $DAS_{30cm}$) to estimate the biomass fraction of smaller trees and data scaled up to 1 ha. Standing dead trees biomass is measured using similar method and not counted in Stem Biomass. | 2013 – 2021 (NXV-01) 2013 – 2020 (NXV-02) | 2013, 2015, 2018 and 2021 (NXV-01) 2013, 2015, 2017 and 2020 (NXV-02) |



| | Biomass of each stem was calculated using Rezende et al. (2006) specific allometric equation for the Cerrado: C = 0.24564+0.01456*(D/10)^2*H where C is aboveground Carbon stocks (kg), D is the diameter (30 cm above the soil), and H is the height (m). The authors assumed that dry stem biomass is 50% carbon. Systematic uncertainty of +25% was assigned to values for error propagation. Errors calculated as the sampling error associated with variation between the transects. | | |
|---|---|---|---|
| Branch turnover net primary productivity ($NPP_{branch\ turnover}$) | Branchfall > 2 cm diameter (excluding that associated with dead trees) surveyed within four 1 m x 100 m transects; small branches were cut to include only the transect-crossing component, removed, and weighed. Larger branches had their dimensions taken (diameter at three points), and assigned a wood density value according to decomposition class (Harmon et al., 1995). Biomass of each branch was calculated. The first collection of branchfall in 2014 lead to 'woody debris biomass', which accounts for nacromass in the ground litter layer. 'Woody debris biomass' is not included in above nor belowground biomass. The biomass of branch has been implicitly included in $NPP_{stem}$. Branchfall was then collected from the same transects every 3 months which lead to $NPP_{branch\ turnover}$ | 2014 – 2019 | Every 3 months |
| Litterfall net primary productivity ($NPP_{litterfall}$) | Litterfall production of dead organic material (< 2 cm diameter) calculated by collecting litterfall in 0.2827 m² circular collectors placed at 1 m above the ground at the center of each of the 25 subplots in each plot. Litter separated into leaves, twigs, reproductive parts (flowers, fruits, and seeds), and unidentifiable. $NPP_{litterfall}$ calculated as follows: $NPP_{litterfall} = NPP_{canopy}$ – Loss to Leaf Herbivory. Litterfall separated into different components, oven-dried at 65°C to constant mass, and weighed. Litter estimated to be 49.2% carbon, based on mean Amazonia values (S. Patiño, unpublished analysis). Errors calculated as the sampling error associated with variation between the litter traps. | Jan 2014 – Dec 2019 | Every 14 days |
| Leaf Area Index (LAI) and Leaf biomass | Hemispherical photos taken with a digital camera (Nikon OP 10mm) and hemispherical lens (Nikon 10mm fisheye lens) near the center of each of the 25 subplots in each plot at a standard height of 1 m and during overcast conditions. LAI estimated from these images using Hemisfer software (licensed version 2.12; http://www.wsl.ch/dienstleistungen/produkte/software/hemisfer/index_EN). LAI estimated from hemispherical photos using the standard Li-Cor LAI-2000 method, based on the Miller (1967) equations, and correcting for non-linearity and slope effects (Schleppi et al., 2007) and canopy clumping (Chen & Cihlar, 1995). Thresholds were set to detect separately for each ring (6 rings) according to Nobis & Hunziker (2005). Errors calculated as the sampling error through variation among subplots. Leaf **biomass** calculated as leaf area index (LAI)/specific leaf area (SLA), where LAI is the plot mean over the study period, and SLA is the basal area-weighted plot mean | Jun 2015 – Jan 2020 | Every 3 months |



| | | | |
|---|---|---|---|
| | over the study period. We used the SLA value of March 2014 (Neyret et al., 2016). | | |
| Loss to leaf herbivory (*NPP*ₕₑᵣᵦᵢᵥₒᵣy) | Estimated based on Neyret et al. (2016)'s observation that the loss to herbivory was 3.11% in NXV-01 and 4.43% in NXV-02. The data collection was conducted between March and May 2014. Each leaf's fractional herbivory ($H$) was calculated as $H = (A_{nh} - A_h) / A_{nh}$. Where $A_h$ is the area of each leaf, including the damage incurred by herbivory, and $A_{nh}$ is the leaf area prior to herbivory (Neyret et al., 2016). The average value of $H$ of all leaves collected per litterfall trap was derived, and plot-level means were calculated. Systematic uncertainty of $\pm50\%$ assigned to values for error propagation. | nc | nc |
| **Below-ground net primary productivity (*NPP*_BG)** | | | |
| Coarse root net primary productivity (*NPP*_coarse root) | Root biomass estimated based on Miranda et al. (2014) that is specific for the vegetation types of Cerrado. Based on this study, the Root(belowground): shoot ratio (aboveground) biomass is 1.37 to *cerrado* and 0.22 to *cerradão*. Systematic uncertainty of $\pm20\%$ assigned to values for error propagation. We used these ratios, 1.37 (at NXV-01) and 0.22 (at NXV-02) to derive *NPP*_coarse root from *NPP*_stem | nc | nc |
| Fine root net primary productivity (*NPP*_fine root) and fine root biomass | In each plot, sixteen ingrowth cores (mesh cages 12 cm diameter, to 30 cm depth) were installed. Roots were manually removed from the soil samples in four 10 min time steps, according to a method that corrects for underestimation of biomass of hard-to-extract roots (Metcalfe et al., 2007) and used to predict root extraction beyond 40 min (up to 120 min); typically, there was an additional 33% correction factor for fine roots not collected within 40 min. Correction for fine roots productivity below 30 cm depth (Galbraith et al., 2013) increased the value by 39%. Errors were calculated as the sampling error associated with variation between the sampling points.<br><br>Root-free soil was then re-inserted into the ingrowth core. Collected roots were thoroughly rinsed, oven-dried at 65°C to constant mass, and weighed. This process was repeated for each measurement thereafter. **Fine root biomass** was calculated from harvested fine roots during the first installation of ingrowth. The subsequent fine root collection from the ingrowth cores lead to *NPP*_fine root | Sep 2014 – Feb 2020 | Every 3 months |



### 2.4 NPP calculation

We measured the NPP in the two plots between 2014 and 2020 (Table 1). We calculated all major components of NPP using the following equations:

$$NPP_{total} = NPP_{coarse\ root} + NPP_{fine\ root} + NPP_{stem} + NPP_{branch} + NPP_{litter\ fall} + NPP_{herbivory} \quad (1)$$

$$NPP_{canopy} = NPP_{litter\ fall} + NPP_{herbivory} \quad (2)$$

$$NPP_{woody} = NPP_{coarse\ root} + NPP_{stem} + NPP_{branch\ turnover} \quad (3)$$

$$NPP_{root} = NPP_{fine\ root} \quad (4)$$





$NPP_{ACW} = NPP_{stem}$ (5)

Our calculations above neglect several small NPP terms, such as NPP lost as volatile

organic emissions ($NPP_{VOC}$), unmeasured litter trapped in the canopy, or litter dropped from
understorey flora below the litter traps (1 m). However, in central Amazonia, Malhi et al. (2009)
found $NPP_{VOC}$ was a relatively minor NPP term (0.13 + 0.06 Mg C ha$^{-1}$ year$^{-1}$). For belowground
NPP, we do not include root exudates and mycorrhizae that account for < 2 Mg C ha$^{-1}$ year$^{-1}$,
representing a modest part of the carbon fluxes (Malhi et al., 2017). Thus, we focus on the canopy,
wood, and fine roots productivity, which account for over 85% of NPP (See Riutta et al., 2018 and
their references).

We calculated the relative allocation to the main NPP components (woody, canopy, and

fine roots NPP) for leaves, fine roots, and stem following the equations:
$Allocation_x = (NPP_x * 100)/NPP_{total}$ (6)

***2.5 Calculation of measurements uncertainty***
Estimation of measurements uncertainty for each NPP component is explained in details in Table
1. For components that are not directly measured, for example $NPP_{total}$ as a sum of several
components, we combine relevant error by error propagation with standard quadrature rules
(Hughes & Hase, 2010; Malhi et al., 2015). During the above process, we also assigned significant
systematic errors to capture uncertainties related to sampling methodology or scaling approaches
(see Table 1); these factors were consistent with those applied in similar previous studies (Malhi
et al., 2009, 2015; 2017; Girardin et al., 2010; Galbraith et al., 2013).

***2.6 Data analyses***
Our analyses were focused on comparing NPP among the years (2014 to 2019) – comprising the
periods before, during, and after the El Niño 2015/2016 events – in *cerrado* and *cerradão*. We
compared the stem and canopy biomass of the two vegetation types over time using repeated
two-way analysis of variance (ANOVA-two way). We used Tukey's post hoc test to compare the
different years in each plot. We used the same analysis to compare productivity and carbon
allocation across different compartments. In cases where the residuals violated the ANOVA
assumptions, we used Friedman's non-parametric analysis. We performed all analyses in the R
environment and adopted a significance level of 0.05. To improve the accessibility of colour
figures with COLORBREWER 2.0.

**3 Results**
***3.1 Net primary productivity***
The net primary productivity (NPP) in the *cerradão* was ~ 30% higher compared to that of the
*cerrado* prior to the occurrence of El Niño (*cerradão* = ~9.3+0.57 Mg C ha$^{-1}$ year$^{-1}$; *cerrado* =
~6.5+1.12 Mg C ha$^{-1}$ year$^{-1}$ Fig. 2; Table S2). This is due to the greater productivity in the canopy
and stem in the *cerradão* (Fig. 2; Table S2). During the El Niño, *cerradão* NPP decreased to





6.6±0.6 Mg C ha⁻¹ year⁻¹ and became similar to the *cerrado* (6.6±1.3 Mg C ha⁻¹ year⁻¹; Fig. 2;
Table S2).
*Cerradão* NPP was severely affected in 2016 during the El Niño event (-29%). In 2018 it
was still 13% lower than pre-El Niño conditions (Fig. 2). Additionally, stem biomass declined
significantly after El Niño (F= 19.3, p< 0.001) and did not return to the values registered before
the event (Fig. S1).
In the *cerrado*, NPP did not vary much before and during the El Niño. However, in 2018,
productivity reduced by ~30%, due especially to the reduction in stem productivity. Despite this,
stem biomass was not significantly influenced by El Niño and increased significantly between
2013 and 2018 (F= 3.1; p<0.05), remaining stable until 2021 (Fig. S1).

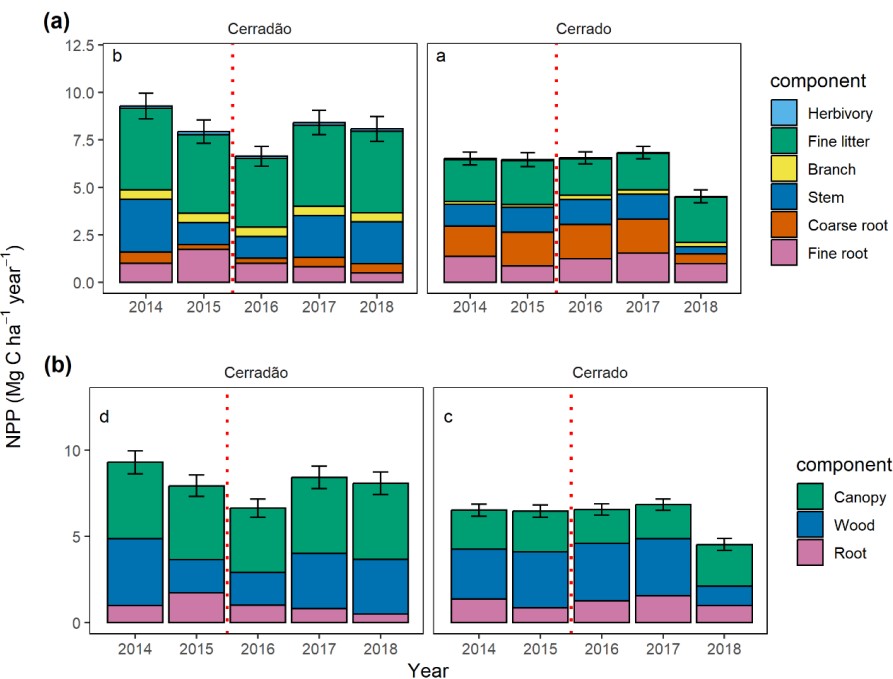


**Fig. 2**. Mean total annual net primary productivity (NPP) between 2014 and 2018 split into its
components (a) and annual NPP allocation into the canopy, wood, and root components (b) at
*cerrado* and *cerradão*. The branch data from *cerradão* was collected in 2014 and repeated in
other years. The error bars represent the standard error for total NPP. The dashed red line
indicates the El Niño periods.


In the *cerradão*, the most affected component was stem net primary productivity (NPPs),
which was reduced by 58% during and after El Niño (F= 15.6, p< 0.001; Fig. 3A). In 2019 it was
still -21% lower than pre-El Niño conditions. When we consider only those trees that were alive
before El Niño and remained alive after the event, the *cerradão* reduced NPPs significantly during
the event, but after the event, NPPs was greater than before the El Niño (Fig. 3B; F= 25.6, p<





0.001). This is mainly due to two critical species for this transitional forest, *Hirtella glandulosa*
Spreng. and *Tachigali vulgaris* L.G.Silva & H.C.Lima, which contributed 22% and 17% to NPPs
after El Niño. Before El Niño, *T. vulgaris* was the species that most contributed to NPPs (26%).
In the *cerrado*, trees showed less diameter growth during and after the event (Fig. 3B; F= 109.7,
p< 0.001). However, stem productivity was not affected during the event (Fig. 3A).

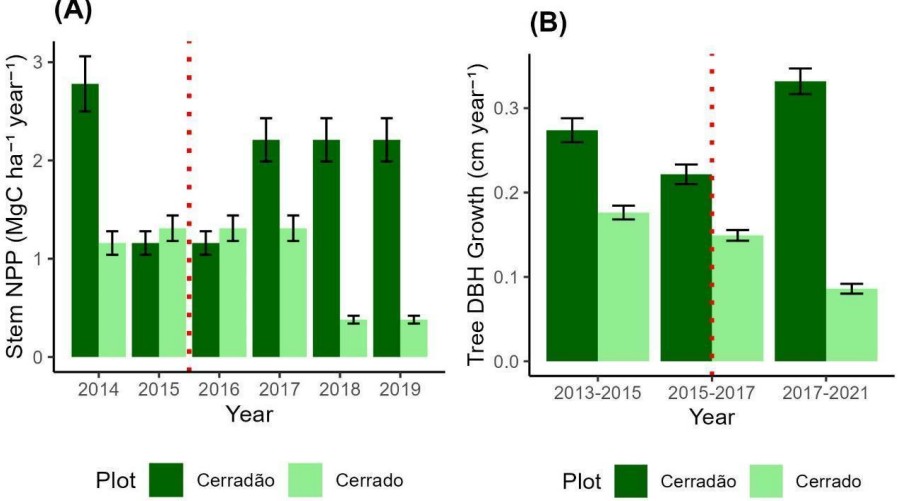


**Fig. 3**. Stem dynamics in *cerrado* (light green) and *cerradão* (dark green). (a) Stem net primary
productivity (NPPstem, MgC ha$^{-1}$ year$^{-1}$) for stem larger than 5cm diameter. We note that there
are negative stem NPP values due to those trees that lose bark or water from the stem in the dry
periods, especially in *cerrado* after 2017. (b) The growth of tree diameter (measured at 30cm
above soil surface) (cm year$^{-1}$), calculated as the increase in DAS between two censuses divided
by time. Only growth is included, in other words, trees with shrinking stems are excluded The
dashed red line indicates the El Niño periods.


In the *cerradão*, fine root net primary productivity (NPPfr) production increased significantly

(+42%) during El Niño (F= 17.3, p< 0.001), but in later years productivity declined (Fig. 4). The
*cerrado* presented the opposite pattern observed in the *cerradão*. NPPfr reduced by 38% during
the event (F= 5.6, p= 0.001; Figs. 2 and 4). However, the NPPfr of this component re-established
itself soon after El Niño, but experienced a decline of ~ 38% in 2018.





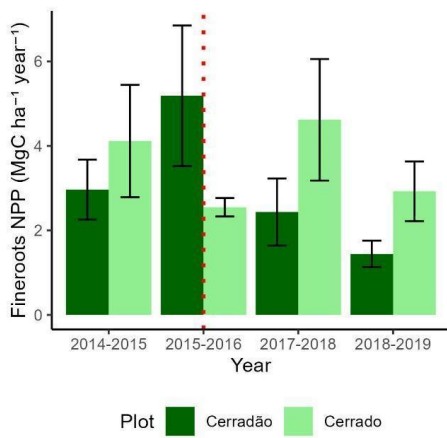


**Fig. 4**. Fine root net primary productivity (NPPfr) for *cerrado* (light green) and *cerradão* (dark green) between September 2014 and August 2019. The error bars represent the standard error. The dashed red line indicates the El Niño periods.



Canopy productivity was affected after the El Niño event in both *cerradão* (F= 2.8, p= 0.01)

and *cerrado* (F= 6.7, p< 0.001) (Fig. 5). However, the NPP of this component was re-established
two years after the event. For the *cerradão*, it is worth highlighting the drop in fruit production after
the event, which had not yet re-established itself two years after El Niño (Fig. 6). Furthermore,
after El Niño, both plots show declining and then recovering LAI (Fig. S2). We also noted that
following El Niño, the variability of LAI increased among subplots, potentially due to clearings
emerging from heightened tree mortality. The average annual mortality rate increased during and
after El Niño, especially in the *cerradão* (Fig. 6).



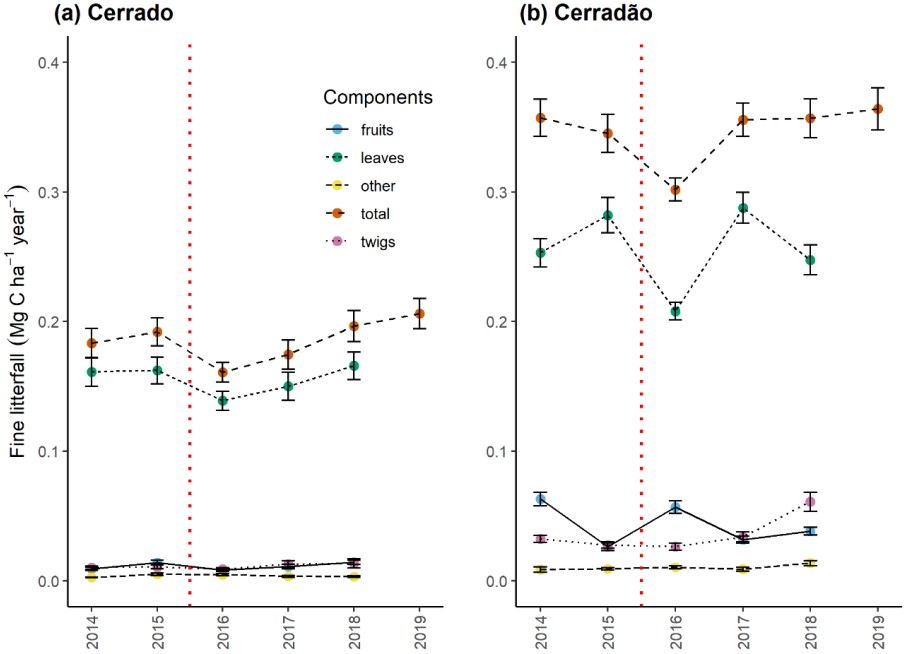


**Fig. 5**. Mean monthly productivity in canopy litterfall and its components for *cerrado* (a) and *cerradão* (b) between 2014 and 2019: (fruits) flower, fruit, and seed fall; (leaves) leaf fall; (other) not identified and (total) total canopy fine litterfall (as measured in litter traps); (twigs) twig fall (< 2 cm). The error bars represent the standard error. The dashed red line indicates the El Niño periods.


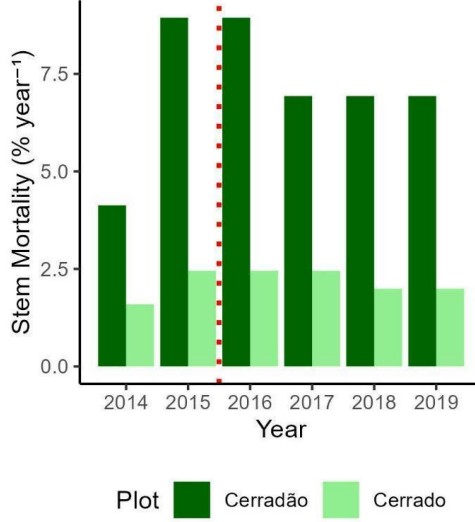






**Fig. 6**. Stem mortality, shown as the percentage of dead trees to the number of trees in the first
census divided by time. The dashed red line indicates the El Niño periods.
***3.2. Allocation between plots, components, and years***
In general, NPP allocation differed significantly between *cerradão* and *cerrado* plots (Fig. 7), but
this varies within the components (F= 41.7, p< 0.001). The allocation to canopy was greater in
*cerradão* (53+3%) than in *cerrado* (37+10%). In contrast, allocation to woody and roots was
greater in the *cerrado* (woody= 44+11%; roots= 19+4%) than in the *cerradão* (woody= 34+8%;
roots= 13+6%). Over the studied time, the NPP allocation showed inter annual variation in *cerrado*
but no clear drought signal the main axis of interannual variation was a trade-off between canopy
investment and root allocation, with woody allocation remaining constant (Fig. 7). However, in
2018, ~ three years after El Niño, the allocation of canopy and wood changed drastically, showing
an opposite pattern to previous years. In the *cerradão*, there is a clear drought signal with
increased investment in fine roots during the drought, and reduced investment in woody growth;
canopy allocation remained fairly constant.




**Fig. 7**. Relative allocation (% of total) of net primary productivity (NPP) to canopy, woody, and
fine root NPP in *cerrado* and *cerradão*. 2014= blue, 2015= pink, 2016= red, 2017=orange and
2018= grey. Woody components include stem, coarse root and branch turnover; Fine root
includes fine root NPP only (no roots exudates); Canopy includes litterfall and herbivory
**4 Discussion**
*Cerradão* and *cerrado* showed contrasting responses to the 2015/2016 El Niño-associated
drought event. The *cerrado* proved to be more resistant and resilient, i.e., most of the parameters





assessed (e.g., stem carbon stock, canopy productivity) changed little during the event, and those
that experienced a reduction soon re-established themselves (e.g., production of fine roots). In
contrast, the *cerradão* showed lower resistance and resilience: stem carbon stock and mortality,
productivity and allocation were affected during and after the El Niño event and even three years
after, most of the parameters assessed were not similar to those observed before the event. Our
findings demonstrate the high sensitivity of the *cerradão* to extreme drought events.

The productivity found in our *cerradão* (9.3±0.57) was similar to that observed in transitional

forests in Africa (9.2 to 13.1; Moore et al., 2018), and some low-fertility humid forest sites in
lowland Amazonia in Colombia and Brazil (~8.1 to 10.3; Aragão et al., 2009; Girardin et al., 2010).
It is markedly greater than observed in seven premontane and montane sites in Peru (~3.9 to 6.4;
Girardin et al., 2010) and lower than observed in lowland tropical forest plots in south-western
Amazonia (15.1 ± 0.8 and 14.2 ± 1.0; Malhi et al., 2015) and nutrient-rich soils forests (17.0±1.4;
Aragão et al., 2009). Nevertheless, the total productivity of *cerradão* was more affected (-29%)
than the Amazonia rainforest (-7.6% to -8.5%) during the El Niño drought of 2015/2016 (Machado-
Silva et al., 2021). Moreover, the reduction in stem productivity was much larger (-58%) than that
estimated for tropical forests as a whole (-8.3% in 1997/1998, and -9% in 2015/2016 (Rifai et al.,
2018). This demonstrates the high sensitivity of this vegetation to climate anomalies.

The higher mortality of *cerradão* may be related to the hydraulic characteristics of the main

species, such as *Tachigali vulgaris*, a pioneer species with stomatal control tending to an
anisohydric condition and, therefore, more susceptible to hydraulic failure (Jankoski et al., 2022).
In addition, many species are brevi-deciduous, so the plants continue to photosynthesize even
during water scarcity (e.g., Jancoski et al., 2022). Other factors, such as a lack of strategies to
avoid water loss, may also contribute, like low trichome density in their leaves and smaller stomata
(Araújo et al., 2021b). Another possible cause of the high mortality in the *cerradão* is the unusually
intense winds that hit trees with xylem tissue already weakened by the effects of drought and heat
due to the El Niño event (Reis et al., 2022). *Cerradão* trees are taller than *cerrado* trees, which
makes them more susceptible to wind disturbances. Once broken, even just part of the crown,
the tree is at greater risk of death (Reis et al., 2022).

Despite the high mortality, the trees that remained alive showed higher stem productivity

than before the El Niño. This may be related to the greater opening of clearings, favouring carbon
uptake and plant growth due to the greater availability of light, water and nutrients to the remaining
trees. During the El Niño drought, a decline in the growth of *Tachigali vulgaris* trees was observed,
leading to a shift in the primary contributor to stem productivity from *T. vulgaris* to *Hirtella*
*glandulosa*. The role reversal of these two species can be explained by the high mortality and low
growth rate of *T. vulgaris* during and after El Niño. *T. vulgaris* is considered a key species for
*cerradão* due to its high biomass gain after disturbances such as fire (Reis et al., 2015; 2017),
but it is sensitive to drought. The high sensitivity of *T. vulgaris* to drought events may be attributed
to the increased xylem tension required to extract water from the soil, making it more prone to
embolism (Jancoski et al., 2022). Consequently, this results in reduced growth and higher
mortality compared to *H. glandulosa*. On the other hand, *H. glandulosa* proved to be more




resistant: it has high foliar trichome density, which works as a strategy to prevent water loss (e.g. Gianoli & González-Teuber, 2005; Araújo et al., 2021b). In the *cerrado*, we observed the opposite pattern; the productivity of trees that remained alive continued to decline after the event. In this vegetation type, plant mortality was low, and the surviving plants had to compete to stay alive, which may explain the lower productivity after El Niño. Furthermore, many trees in the *cerrado* shed their outer bark, which may have affected the diameter measurement and, consequently, the productivity of the stem. The loss of bark from *cerrado* plants, especially after fire and drought events, makes the measurement of trunk productivity inaccurate.

The high production of fine roots in *cerradão* during drought is probably a strategy to increase soil water uptake during the period of soil water scarcity (Metcalfe et al., 2008). However, this strategy on partially ameliorates drought risk, as tree mortality was high even with a high investment in fine roots. The *cerrado*, on the other hand, showed the opposite strategy, investing less in fine roots during the event. However, shortly afterwards, the productivity of this component was similar to that observed before El Niño. Lowland *terra firme* have less root growth during the dry season but had greater specific root length and surface area where soil moisture was depleted (Metcalfe et al., 2008), and the *cerradão* presented a strategy similar to these Amazonia forests, but not the *cerrado*.

Both *cerrado* and *cerradão* adopted the strategy of losing more leaves during El Niño. It is well known that during periods of water stress in the soil, plants lose their leaves as a strategy to avoid water loss and consequent mortality (e.g., Brando et al., 2008). This strategy can also have nutrient cycling benefits: the nutrients released to the litter layer and soil after leaf drop and can later be reabsorbed by the plants when they re-establish leaf growth after a high stress period (e.g., Oliveira et al., 2017). The high leaf loss during El Niño may have contributed to lower photosynthetic activity of plants (e.g., Luo et al., 2018; Kaewthongrach et al., 2020), consequently affecting carbon accumulation.

The canopy-wood-fine root trade-offs identified here are different from those reported by Doughty et al. (2014) for a somewhat similar Amazonian forest-dry forest transition in Bolivia, with similar rainfall but more fertile soils. There, the site with better water availability (related to soil properties) hosted an Amazonian forest which showed wood-canopy trade-offs during drought. The drier site hosted *chiquitano* dry forest with wood-fine root trade-offs during drought. Our *cerradão* site shows similar wood-fine root trade-offs to the *chiquitano* forest, whereas our *cerrado* site shows a different canopy-fine root trade off. One possibility is that these shifting trade-off strategies reflect points on an aridity continuum from sub-humid Amazonian forest (wood-canopy trade-off) through transitional or seasonally dry forests (wood-fine root trade-off) through to savanna (wood-canopy trade-off). Alternatively, the differences in soil fertility may play a role, changing the costs and advantages of investment in fine-root production.



**5 Conclusions**

*Cerradão* is a vital transitional vegetation type at the Amazon-Cerrado ecotone, as it is in contact with the two main Brazilian biomes, Cerrado and Amazonia. However, this vegetation type appears to be susceptible to climatic events (*present study*), wildfires (Reis et al., 2015; 2017) and wind storms (Reis et al., 2022). One of the most frequent species in *cerradão* (*T. vulgaris*), that is especially important for carbon uptake, proved to be very sensitive to the climatic event. Thus, if these extreme drought events continue to become more frequent and intense, cerradão may release more carbon than absorbs, as observed here. In addition, the *cerradão* serves as a connection between the savanna and the forest, acting as a kind of buffer-barrier for the Amazonia to the effects of environmental stressors along its contact with the *cerrado*. Our results suggest that the more frequent occurrence of El Niño events can break this natural barrier, creating conditions for the progressive degradation of the forest along the edges.

**Author Contributions**: S.M.R. wrote the manuscript with input from all authors (Y.M., B.H.M.Jr., R.F., B.S.M., H.Z., C.A.J.G., E.A.O., K.S.P., L.J.S., E.L.S., E.B.S., K.P.S., M.D.A.G., C.A.L.D., O.L.P. and I.O.M.); Y.M., B.H.M.Jr. and I.O.M. were involved in planning and supervised the work; S.M.R., R.F., E.A.O., K.S.P., L.J.S., E.L.S., E.B.S., K.P.S. and M.D.A.G. performed the field measurements; S.M.R., H.Z., and C.A.J.G. performed the analyses and made the figures. All authors discussed the results and contributed to the final manuscript.

**Data availability statement:** The data are available at Dryed:
https://doi.org/10.5061/dryad.rjdfn2zhw (Reis, 2023).

**Funding statement:** National Council for Scientific and Technological Development (CNPq) - financial support of the projects PELD "Cerrado-Amazonia Transition: ecological and socio-environmental bases for Conservation" (stages II, III and IV) - 403725/2012-7, 441244/2016-5 and 441572/2020-0, PPBIO "Phytogeography of the Amazon-Cerrado Transition Zone" (457602/2012-0) and FAPEMAT (164131/2013 and 0589267/2016). S.M.R. was funded by a postdoctoral Fellowship from NERC and FAPESP (BIO-RED 2015/50517-5). This paper is a product of the Global Ecosystems Monitoring (GEM) Network, which was supported by an ERC Advanced Investigator Award to YM.

**Acknowledgements**: We thank the team of the Laboratório de Ecologia Vegetal - Plant Ecology Laboratory at the UNEMAT (Universidade do Estado de Mato Grosso) campus in Nova Xavantina, especially to Carla Heloísa Luz de Oliveira, Camila Borges, Erica Prestes Ferreira, Luiz Macedo Schuwaab Júnior, Erika Camila Oliveira, Izabel Amorim, Eder Carvalho das Neves, Kelen Alves Cavalheiro and Poliana Alves Cavalheiro for help collecting field data. We also thank the National Council for Scientific and Technological Development (CNPq) for financial support of the projects PELD "Cerrado-Amazonia Transition: ecological and socio-environmental bases for Conservation" (stages II, III and IV) - 403725/2012-7, 441244/2016-5 and 441572/2020-0, PPBIO "Phytogeography of the Amazon-Cerrado Transition Zone" (457602/2012-0) and



FAPEMAT (164131/2013 and 0589267/2016). We also thank CNPq for research productivity
grants PQ1 to B.S. Marimon and B.H. Marimon Junior. S.M.R. was funded by a postdoctoral
Fellowship from NERC and FAPESP (BIO-RED 2015/50517-5). This paper is a product of the
Global Ecosystems Monitoring (GEM) Network, which was supported by an ERC Advanced
Investigator Award to YM.

**Competing interests**
The contact author has declared that none of the authors has any competing interests.

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
