# Peer review of "Sensitivity of tropical woodland savannas to El Niño droughts"

_EGUsphere, 2024_

## Author Comment (AC1)

Sensitivity of tropical woodland savannas to El Niño droughts

*Originality and significance*

The manuscript titled "Sensitivity of tropical woodland savannas to El Niño droughts" provides an ecosystem-level analysis of the effects of the strongest El Niño event on record on the biomass productivity of two vegetation types (Cerrado and Cerradão) in the Brazilian Cerrado. The study synthesizes approximately eight years of productivity data, some of which were collected at an intra-annual frequency. This research is valuable as it expands our understanding of a critically important and endangered biome in South America.

Response: Thank you for your thoughtful and positive feedback. We have carefully considered the comments raised in this review and have made revisions throughout our manuscript to address each point.

*General comments*

Overall, the manuscript is well-written, though certain sections could benefit from further clarification, and organizing content into subsections may help enhance readability. For example, the discussion is currently presented as a single section that addresses multiple aspects of ecosystem productivity: stems, canopy, fine root production, biomass allocation, and ecosystem-level NPP. The authors might consider restructuring these sections to better highlight their results. This suggestion also applies to the results section. Additionally, some passages could be refined to improve clarity, and certain sentences might benefit from adjustments in English style and grammar to enhance overall readability. Some of my general comments include:

Response: Thank you very much for your positive and constructive feedback. We appreciate your suggestions on improving clarity and readability by introducing subsections within the results and discussion sections. We agree that this restructuring, particularly within the discussion on ecosystem productivity, will help better highlight key findings and make the manuscript more accessible.

In line with your recommendation, we will also refine certain passages and make adjustments in style and grammar to enhance overall readability. We are confident that these revisions will significantly improve the manuscript, and we are grateful for your guidance in achieving this.

- **Introduction & main hypothesis:** What are the expected differences in how these plant communities respond to ENSO events, and what are the underlying reasons for these differences? The introduction currently lacks sufficient information to propose a well-developed hypothesis in this direction. It mainly

describes basic characteristics of the plant communities without clearly linking this information to the research objectives, making it difficult for the reader to grasp the relevance of these details. A critical missing element is a discussion on biomass allocation patterns in response to drought and their significance for productivity in the Cerrado, which is essential given that much of the discussion revolves around these patterns. Additionally, a more nuanced description of physiological differences beyond those briefly mentioned in lines 85-96 could better illustrate how these differences influence sensitivity to climate change, providing stronger context for the study's hypotheses. Addressing these gaps would require a comprehensive revision and restructuring of the introduction, including clearer research questions. Breaking down the introduction, results, and discussion into distinct sections could also improve the flow and clarity of the manuscript.

Response: Thank you very much for your detailed suggestions. We will restructure the introduction to provide a more comprehensive overview of biomass allocation patterns, particularly in relation to drought conditions and productivity in the Cerrado. We recognise the importance of these patterns for understanding the research findings and will aim to highlight their significance more effectively. Furthermore, we will expand on the physiological differences between plant communities, going beyond the initial description in lines 85-96, to better contextualise the anticipated sensitivities of these communities to climate change.

We note, however, that there is currently very limited information available on the effects of ENSO specifically on the Cerrado, which restricts our ability to elaborate in this area fully. Nevertheless, we will incorporate as much relevant information as possible to strengthen the context of our hypotheses.

- **NPP estimated metrics:** It is unclear why the authors estimate the contribution of herbivory to NPP and coarse root production, as these estimates are not used in any analysis or discussion. Including these estimates adds confusion and does not contribute significantly to the overall narrative, especially since these fluxes were not directly measured. Inferring their sensitivity to ENSO based on estimations with prescribed uncertainty could be highly misleading and does not add anything to the manuscript.

Response: Thank you very much for your comments on the NPP estimated metrics. We understand your concerns regarding the inclusion of herbivory and coarse root production estimates, especially given that these fluxes were not directly measured and are not used in the analysis or discussion.

We included these estimates to avoid underestimating NPP, following common practice in similar studies (as noted in GEM network publications). Including this component allows for a more accurate and comparable NPP estimate with other studies. Additionally, these estimates are scaled from other measured components and, as such, do not influence the interpretation of the interannual variation in NPP.

In light of your feedback, we will ensure that this rationale is clearly stated in the manuscript to avoid any potential confusion. Thank you once again for highlighting this, as we believe these clarifications will strengthen the clarity of our approach.

- **Cerrado 2018 productivity data:** The 2018 productivity data for the Cerrado is not clearly explained, and the authors only briefly touch upon the reasons for the observed decline. Why was productivity so much lower in 2018? The authors suggest a lack of accuracy in the diameter readings (lines 367-370), raising concerns about the reliability of the data. This also prompts the question: was the data prior to 2018 accurate, or could similar measurement errors have led to overestimations and potentially flawed conclusions?

Response: Thank you for your observations regarding the 2018 Cerrado productivity data and the potential sources of error in diameter measurements. We understand your concerns about the reliability of this dataset and the need for a clearer explanation of the decline observed in that year.

The significant decrease in productivity in 2018 is likely due to a combination of environmental factors and natural variation in individual growth responses within the plant community, rather than solely issues with measurement accuracy. While some stems may exhibit bark shedding, we do not believe this to be a major source of error in the diameter measurements for this dataset. The variability observed is more likely to reflect genuine differences in growth responses among individual plants.

We would like to emphasise that the data collected throughout the entire experiment, including prior to 2018, are considered reliable. We followed consistent quality control procedures and applied standardised protocols to all measurements, ensuring data accuracy across the study period.

Nevertheless, we will verify the 2018 data by cross-checking the diameter measurements. Additionally, we have quarterly diameter measurements for 200 trees, which will allow us to reassess the accuracy of the 2018 data. This re-evaluation will help us confirm the robustness of our findings and ensure transparency in our analyses and conclusions.

- **Statistical analysis:** This section lacks critical information. The statistical models used for each analysis are not clearly described, leaving the reader uncertain about which model was applied to address specific research questions. This is particularly important because the data originate from only two plots (n=1 per forest type), raising questions about the sample size and analysis approach. Did the authors use subplots, litterfall traps, or individual trees as sampling units instead? If so, how was spatial autocorrelation within subplots accounted for? None of these methodological details are explained in the statistical analysis section, making it difficult to assess the robustness of the results.

Response: Thank you for your comment. We appreciate the opportunity to clarify the statistical approach used in this study.

For the comparison of stem biomass, we used subplots within each forest type as the sampling units. For canopy biomass, we relied on litterfall collectors. The plots were systematically distributed in a contiguous manner, while the litterfall collectors were randomly placed within each 1-hectare plot to ensure representative sampling.

In our statistical models, we included subplots and litterfall collectors as random effects to account for spatial variability within the plots. Additionally, we tested for interactions between plot and year to address temporal variability. This modeling approach allowed us to address the inherent variability in the data while making full use of the available sampling units.

We will revise the manuscript to include these methodological details in the statistical analysis section, ensuring that the description is clear and transparent for readers.

- **Figures:** The figures need several enhancements to improve clarity. For example, pairwise comparisons are not indicated, making it difficult for readers to discern if values are significantly different. Some figures, like Figure 1, have font sizes that are too small, and others, like Figure 6, are missing error bars. In Figure 2, letters appear within the panels (upper left corner) that are not explained in the figure legend, leading to confusion. Although the manuscript specifies that the color scheme was chosen to be accessible to color-blind readers (lines ), many bar plots still use similar shades of green and include a dashed red line, which may not be effective. A simpler color scheme, such as using white and dark gray/black, might improve readability.

Response: Thank you for your comments regarding the figures. We agree that some improvements are needed to enhance clarity and facilitate data interpretation. Below are the actions we will take to address your suggestions:

Regarding pairwise comparisons, statistical differences will be indicated in the figures, with significant results clearly marked. However, including too many letters on the figures may hinder readability. If this becomes an issue, we will provide a supplementary table with the detailed pairwise comparisons for the readers' reference.

Regarding font sizes in figures, we will adjust the font size in the figures, particularly in Figure 1, to ensure all text is legible.

For Figure 6, we calculated the mortality rate for the plot, which is why no error bars are included. We will clarify this in the text.

The letters appearing in the panels of Figure 2 are not correct, and we will remove these letters to improve clarity.

Regarding colour scheme, we will revise the colour scheme to ensure it is accessible to colour-blind readers. Although we initially chose a specific scheme for this purpose, we will now adjust the bar colours to use higher contrast combinations.

*Line-by-line comments below:*

Lines 25-26: Perhaps you could also highlight whether these differences persisted during ENSO.

Response: Okay. We will highlight whether these differences persisted during ENSO events.

Line 42: I think the neotropics in general experience the strongest drought on record. For instance tropical dry forest in Central America suffer exacerbated mortality rates, to the point that up 30% of all individuals of a given species died (Powers et al., 2020).

Response: We will include this information in the manuscript to provide a broader context for the significance of drought impacts on ecosystem productivity and mortality in the region.

Line 66: This idea repeats one item (wildfires) in the list above. Maybe delete from the list and just state it here.

Response: We agree and will exclude wildfires from the list above, addressing it directly in the following sentence instead.

Line 89: Perhaps you can use the word "indicates" instead of "means".

Response: We will replace the word "means" with "ensures" in line 89 to improve the clarity and precision of the language.

Lines 89-90: Is this a hypothesis you are proposing? One could argue that the earlier the leaf loss is due to more hydraulic vulnerability in these species (Reich & Borchert, 1984; Eamus, 1999; Brodribb et al., 2002; Sobrado, 2015; Vargas G. et al., 2021).

Response: Thank you for your comment. To clarify, this is not a hypothesis but rather a finding reported by Araújo et al. (2021b). In this study, we are examining the same species co-occurring at both sites to investigate their responses under different environmental conditions. We will revise the text to make this distinction clearer.

Line 93: tree height and plant vulnerability to drought. Maybe you would like to check the work by Olson and collaborators (2018, 2020).

Response: Thank you for your suggestion. We will review the work by Olson and collaborators (2018, 2020) to incorporate relevant insights into our text.

Additionally, the work we previously cited as "under revie " has now been published: Araújo, Igor; Marimon, Beatriz S.; Marimon Junior, Ben Hur; Oliveira, Carla H.L.; Silva, José W.S.; Beú, Raiane G.; da Silva, Ivone Vieira; Simioni, Priscila F.; Tavares, Júlia V.; Phillips, Oliver L.; Gloor, Manuel U.; Galbraith, David R. (2024). Taller trees exhibit greater hydraulic vulnerability in southern Amazonian forests. Environmental and Experimental Botany, 226, 105905.

Line 97: Maybe state the main hypothesis of the study here. Also, what are the broader implications of studying these responses. Can they inform how we predict the fate of this ecosystem in a warmer/drier world.

Response: Thank you for the suggestion. We agree that stating the main hypotheses or expectations of the study at this point would strengthen the introduction. We will revise the text to include our hypotheses, focusing on the effects of the El Niño event on productivity and biomass allocation.

However, we believe that discussing the broader implications, such as the fate of this ecosystem in a warmer or drier world, is more appropriate for the Discussion section. There are limitations to extrapolating short-term changes to long-term responses, and we will address these in the context of our findings in the Discussion.

Line 108: What is the size of the plots, how many subplots are there? Something missing in the site description is the water table proximity, which is really important in this region of the world (Mattos et al., 2023).

Response: Thank you for your comment. We will provide additional details about the plot size and the number of subplots in the revised manuscript to clarify this aspect.

Lines 157-159: Are you referring here to the hydrological year? Or is just the period being there was ENSO? I think using the hydrological year is the most appropriate (Aragão et al., 2007; Feng et al., 2013; Schwartz et al., 2020).

Response: Thank you for your comment. We are using the hydrological year in this section, as defined by Aragão et al. (2007). We will revise the text to clarify this point and ensure consistency with the reference to the hydrological year.

Table 1: the writing in some of the descriptions is confusing (e.g., LAI).

Response: We will review and revise the descriptions, including those related to LAI, to improve clarity and readability.

Table 1: Loss to leaf herbivory and coarse root net primary productivity. Is the 1.37 the average of "shrubland" in Miranda et al. 2014? Since this was not measured and the systematic uncertainty seems arbitrary and do not follow the CV presented by Miranda et al. 2014, I would restrain of using this.

Response: Thank you for your comment. The value of 1.37 is indeed based on the average for "shrubland" from Miranda et al. (2014). We acknowledge the limitations of applying this estimate and the associated uncertainty. However, we believe these components are valuable for estimating total NPP, as they provide a more complete picture of ecosystem productivity. While estimating total NPP is not the core aim of this paper, we find it useful to include this information, given the scarcity of such estimates for savannas.

Additionally, we note that a recent study by Terra et al. (2023) found a similar relationship, with a value of 1.58 for belowground productivity in Brazilian savannas. We will incorporate this reference to provide further context and support for the inclusion of these estimates in our discussion.

Lines 182-198: I would focus the estimations on the measurements you have data for. The problem of adding estimations with systematic errors, is that these can be highly inaccurate and can lead to misleading results. While I appreciate the intend to be as precise as possible in estimating the NPP. The focus of the study is to quantify the

effects of ENSO on the measured fluxes. Adding all this information on unmeasured quantities is just confusing and does not seem to add much to the study.

Response: Thank you for your insightful comment. We understand your concern regarding the inclusion of estimations for unmeasured quantities and the potential for systematic errors to introduce inaccuracies. However, as mentioned in a previous response, we believe these components are valuable for estimating total NPP, as they provide a more comprehensive picture of ecosystem productivity.

Line 188: This is just the production of fine roots, because you are not considering the production of coarse roots.

Response: We agree with your observation and we will exclude this equation.

Lines 207-210: Maybe is worth explaining here the decision making process to define these % of error propagation.

Response: Thank you for your comment. These values were, to some extent, conservative expert judgements based on our familiarity with the methods and potential systematic issues. We aimed to strike a balance between acknowledging uncertainties and providing reliable estimates. We will add a more detailed explanation of this process in the revised manuscript to ensure transparency and improve the readers' understanding of our approach.

Lines 215-216: Please describe the models more. Main effects? nested effects? Interactions? Also, how did you deal with a sample size of 1 per forest type?

Response: Thank you for your comment. To clarify, for the comparison of stem biomass, we used the subplots within each forest type, while for canopy biomass, we used the litterfall collectors. The plots are systematically distributed in a contiguous manner, and the litterfall collectors are randomly placed within each 1-hectare plot.

We included the subplots and collectors as random effects in the models and tested for interactions between plot and year. This approach allowed us to account for spatial and temporal variability effectively. We will revise the manuscript to make these methodological details clearer for the reader.

Lines 220-221: This is unnecessary, and shades of green with red dashed lines is not a color-blind friendly combination of colors.

Response: We agree with your observation and will exclude this information from the manuscript.

Line 229: Can we say then that ENSO did not affect the Cerrado?

Response: Yes, we can say that ENSO did not affect the overall productivity of the cerrado. This information was in the paragraphs below, but now I have included it in the same paragraph.

Line 240: What are these little letters in the upper left corners, what do they represent?

Response: We believe these elements are unnecessary and propose their exclusion.

Lines 250-253: This is not correct. First, 3 b) shows dbh growth, not NPP of alive trees. Second, npp should consider the biomass lost to mortality (Anderson-Teixeira et al., 2016).

Response: Thank you for your suggestion. We agree and will exclude it to avoid any potential confusion.

Line 260: Can you indicate whether you observed differences in the plot?

Response: Yes, we can proceed with that.

Line 276: Again, indicate with letters pairwise comparisons.

Response: Yes, we can proceed with that.

Line 291: Indicate significance here.

Response: We can include letters to indicate significance; however, the figure may become overly cluttered. It might be more appropriate to include a table in the supplementary material instead.

Lines 310-312: This is not clear from the figures.

Response: We changed the text and also included more information. We have also included the allocation text along with the total productivity (Figure 2) as suggested by reviewer 2 and we believe it is now easier to understand.

Line 318: Add the color scale to the plot as a legend, the axes should be black and same with the gridlines.

Response: We agree with your suggestion and will implement the necessary changes.

Line 325: This is not clear from the figures. Resistant, yes when comparing absolute values. Did you try looking at the relative change in productivity? Resilient, is impossible to compare when the 2018 data seem wrong.

Response: Thank you for your insightful comment. We understand that the figures may not clearly support the interpretation, and we appreciate your suggestion to consider relative changes in productivity. We have analysed the relative changes, and it is evident that stem NPP was affected in the cerradão but not in the cerrado. We will revise the text to clarify this finding and improve the interpretation of the results.

Additionally, we will verify the 2018 data by checking the diameter measurements. We also have quarterly diameter measurements for 200 trees, which will help us assess the accuracy of these data. This re-evaluation will ensure that our analyses and conclusions are robust and transparent.

Lines 335-338: This is unnecessary, and just fill space.

Response: Thank you for your comment. While we understand your concern, we believe that the comparison is important as it provides a broader context for our findings, offering a global perspective on the observed values. This comparison helps to position our study within the wider body of literature and highlights how our results relate to similar studies in different tropical forest regions. However, to address your

point, we have summarised the text to make it more concise, while retaining the essential comparison. We hope this revision improves the clarity and focus of the manuscript.

Lines 343-344: dominant species or most important species might be the most appropriate term.

Response: We cahanged to "dominant".

Line 345: plants can adjust this... See Guo et al (2020) (https://doi.org/10.1111/nph.16196) and Guo et al. (2024) (https://doi.org/10.1111/nph.19805). Perhaps is not a matter of adjusting their stomatal control, but more about their hydraulic safety margin? Nowhere in the discussion or introduction hydraulic safety margins are mentioned. Also, in Jankoski et al. 2022, Tachigali vulgaris seems to be partially isohydric (Fig. 3), which suggests some degree of stomatal regulation.

Response: Thank you for your insightful comment. We will revise the sentence to incorporate your suggestion, emphasising the role of hydraulic safety margins (HSM). Additionally, we will refer to the HSM50 value from Jankoski et al. (under review), based on her doctoral thesis. This value indicates that T. vulgaris operates within a critical negative safety margin (-0.2), supporting the idea of hydraulic constraints rather than relying solely on stomatal control.

Lines 349-351: How does the amount of embolisms link to wind resistance? In your previous article (Reis et al. 2022), you assessed mortality but did not gather enough information to suggest that drough+heat decreased the capacity of stems to resist winds. Please explain?

Response: Thank you for your comment. We have revised the text to address your concern. The reference to weakened xylem tissue has been removed, as we acknowledge that the data presented in Reis et al. (2022) did not provide direct evidence to support this claim. The revised text now focuses on the role of wind intensity and tree structure in explaining the high mortality observed in the cerradão. We believe this change makes the discussion more accurate and aligned with the available evidence.

Lines 354-355: higher productivity when?

Response: We have revised the sentence to clarify that the higher stem productivity refers to the period after the El Niño event.

Lines 362-364: This is not clear from that reference....

Response: We will review the reference and revise the text to ensure it aligns more accurately with the findings of the cited study.

Lines 367-370: plants were competing also before ENSO. This statement does not make sense. Was there a fire event after or during ENSO???? The one problem is the magnitude of the inaccuracy, that is concerning.

Response: Thank you for your comment. We agree that competition among plants was also occurring prior to the El Niño event, and we will revise the explanation to clarify this point.

Regarding fire events, the area did not experience any fires during or after the El Niño event. The last recorded fire in the area occurred in 2008.

As mentioned in a previous response, we will also verify the accuracy of the stem NPP data to address concerns about potential inaccuracies and ensure the robustness of our findings.

Lines 377-378: This could also be root phenological patterns associated to water availability (Kummerow et al., 1990; Kavanagh & Kellman, 1992).

Response: Thank you for your suggestion.We included the text: "The strategy observed in the cerrado was similar to that of tropical dry forests, reflecting root phenological patterns linked to water availability (Kummerow et al., 1990; Kavanagh & Kellman, 1992)."

Lines 388-390: Yes, but this was not measured. Something the authors have not developed is whether plants are deciduous or evergreen or brevideciduous and how the underlying physiological differences link to observed responses. There are only a couple of lines about that, but that might have implications in the different responses of the two plant communities. For example:

- It might be important in the regulation of water transport and photosynthesis (Brodribb et al., 2002).

- It might also be related with rooting depth (Smith-Martin et al., 2019, 2020).

Response: Thank you for your valuable comment. We acknowledge that we did not collect data on the deciduousness of the species, which limits our ability to fully explore the role of leaf phenology in the observed responses. Our intention in discussing leaf loss during the El Niño year was to highlight its potential contribution to reduced productivity, as there is a well-established relationship between leaf area and plant growth (Zhang et al., 2014). We will clarify this in the revised manuscript, emphasizing that the increased leaf loss may have impacted the productivity of the plants.

Line 396: this terminology (wood-fine root trade-offs) is confusing as is not defined. The manuscript will benefit from better staging of these concepts in the introduction.

Response: Thank you for your comment. We will revise the manuscript to define this concept more clearly in the introduction and provide a better framework for understanding it. This will ensure that the term is introduced with adequate context and will improve the overall clarity of the manuscript.

Lines 397-400: This is an interesting proposition, but it will benefit from more background information. Basically, can we expect different biomass allocation patterns among biomes?

Response: Thank you for your comment. We agree that this proposition could benefit from additional background information. To strengthen this point, we will expand on the concept of biomass allocation patterns across different biomes.

Line 407: Maybe "important" is more appropriate than "vital".

Response: Done.

Line 411: Are you referring to ENSO or drought?

Response: We are referring to ENSO. The text has been revised to clarify this.

Line 413: this was not quantified.

Response: We have revised the sentence to include the phrase "as observed in a recent study in the region (Gatti et al., 2021)".

Lines 414-415: Why is this? It just appear out of the bleu.

Response: Thank you for your observation. We have revised the text to clarify the role of the *cerradão* as a transitional vegetation type within the Amazon-Cerrado ecotone.

*References*

Anderson-Teixeira KJ, Wang MMH, McGarvey JC, LeBauer DS. 2016. Carbon dynamics of mature and regrowth tropical forests derived from a pantropical database (TropForC-db). Global Change Biology 22: 1690–1709.

Aragão LEOC, Malhi Y, Roman-Cuesta RM, Saatchi S, Anderson LO, Shimabukuro YE. 2007. Spatial patterns and fire response of recent Amazonian droughts. Geophysical Research Letters 34.

Brodribb TJ, Holbrook NM, Gutiérrez MV. 2002. Hydraulic and photosynthetic co-ordination in seasonally dry tropical forest trees. Plant, Cell & Environment 25: 1435–1444.

Eamus D. 1999. Ecophysiological traits of deciduous and evergreen woody species in the seasonally dry tropics. Trends in Ecology & Evolution 14: 11–16.

Feng X, Porporato A, Rodriguez-Iturbe I. 2013. Changes in rainfall seasonality in the tropics. Nature Climate Change 3: 811–815.

Kavanagh T, Kellman M. 1992. Seasonal Pattern of Fine Root Proliferation in a Tropical Dry Forest. Biotropica 24: 157.

Kummerow J, Castillanos J, Maas M, Larigauderie A. 1990. Production of fine roots and the seasonality of their growth in a Mexican deciduous dry forest. Vegetatio 90: 73–80.

Mattos CRC, Mazzochini GG, Rius BF, Penha D, Giacomin LL, Flores BM, Silva MC, Xavier RO, Nehemy MF, Petroni AR, et al. 2023. Rainfall and topographic position determine tree embolism resistance in Amazônia and Cerrado sites. Environmental Research Letters 18: 114009.

Olson ME, Anfodillo T, Rosell JA, Martínez-Méndez N. 2020. Across climates and species, higher vapour pressure deficit is associated with wider vessels for plants of the same height. Plant, Cell & Environment 43: 3068–3080.

Olson ME, Soriano D, Rosell JA, Anfodillo T, Donoghue MJ, Edwards EJ, León-Gómez C, Dawson T, Camarero Martínez JJ, Castorena M, et al. 2018. Plant height and hydraulic vulnerability to drought and cold. Proceedings of the National Academy of Sciences 115: 7551–7556.

Powers JS, Vargas G. G, Brodribb TJ, Schwartz NB, Pérez-Aviles D, Smith-Martin CM, Becknell JM, Aureli F, Blanco R, Calderón-Morales E, et al. 2020. A catastrophic tropical drought kills hydraulically vulnerable tree species. Global Change Biology 26: 3122–3133.

Reich PB, Borchert R. 1984. Water Stress and Tree Phenology in a Tropical Dry Forest in the Lowlands of Costa Rica. Journal of Ecology 72: 61–74.

Schwartz NB, Lintner BR, Feng X, Powers JS. 2020. Beyond MAP: A guide to dimensions of rainfall variability for tropical ecology. Biotropica 52: 1319–1332.

Smith-Martin CM, Bastos CL, Lopez OR, Powers JS, Schnitzer SA. 2019. Effects of dry-season irrigation on leaf physiology and biomass allocation in tropical lianas and trees. Ecology 100: e02827.

Smith-Martin CM, Xu X, Medvigy D, Schnitzer SA, Powers JS. 2020. Allometric scaling laws linking biomass and rooting depth vary across ontogeny and functional groups in tropical dry forest lianas and trees. New Phytologist 226: 714–726.

Sobrado MA. 2015. Leaf Tissue Water Relations Are Associated with Drought-Induced Leaf Shedding in Tropical Montane Habitats. American Journal of Plant Sciences 6: 2128–2135.

Vargas G. G, Brodribb TJ, Dupuy JM, González-M. R, Hulshof CM, Medvigy D, Allerton TAP, Pizano C, Salgado-Negret B, Schwartz NB, et al. 2021. Beyond leaf habit: generalities in plant function across 97 tropical dry forest tree species. New Phytologist 232: 148–161.

###################################################################

Referee #2

In this manuscript, the authors examined how the vegetation in the Cerrado responded to the 2015-16 El Niño event, using field measurements collected between 2014 and 2019. They discovered contrasting patterns between the cerrado and cerradão ecosystems. The dataset employed in this study is particularly valuable and unique, with intriguing results. Nevertheless, there are two major issues that the authors need to address during the revision.

Response: Thank you for your positive feedback. We will carefully consider the comments raised in this review and will make revisions throughout our manuscript to address each point.

1. The 'Results' and 'Discussion' sections must be better structured. At present, the authors present all the information without using sub-headings, particularly in the 'Discussion' section.

Below are the suggestions for the structure of the 'Results' section.

3.1 Total NPP and its allocation

3.2 Canopy NPP

3.3 Stem NPP and mortality

3.4 Root NPP

3.5 Dynamics among canopy, stem and root NPP

The 'Discussion' should adopt a similar structure.

Response: Thank you very much for your constructive comment. We will revise the Results and Discussion sections to organise them into thematic topics, as well as restructure the Materials and Methods section. We believe these changes will enhance the clarity and readability of the manuscript.

2. The authors defined the 12 months from May 2015 to April 2016 as the 2015-16 El Niño event (lines 157-158) and presented temperature, precipitation, and MCWD anomalies during this period compared to other years (Fig. 1). Why not use this same May-to-April definition for the remaining figures in the study? This would provide a clearer understanding of the 2015-16 El Niño event's impact on vegetation.

Response: Thank you very much for your comment. Some variables, such as fine root and branch NPP, are measured quarterly, while others, such as stem NPP and stem mortality, are measured annually. This makes it challenging to calculate NPP for the period from May to April. On the other hand, variables like litterfall are measured monthly, enabling such calculations. We have analysed the data both on an annual basis

and using a May-to-April calendar year. However, the results were similar because plants typically take time to respond to climatic phenomena.

We chose to present the data on an annual basis because adopting a May-to-April timeframe would result in the loss of one year of data: four months from the first year (before May) and eight months from the final year (after April). Since the results are comparable, losing an entire year of data collection would not be advantageous.

---

## Author Response (AR1)

Sensitivity of tropical woodland savannas to El Niño droughts

*Originality and significance*

The manuscript titled "Sensitivity of tropical woodland savannas to El Niño droughts" provides an ecosystem-level analysis of the effects of the strongest El Niño event on record on the biomass productivity of two vegetation types (Cerrado and Cerradão) in the Brazilian Cerrado. The study synthesizes approximately eight years of productivity data, some of which were collected at an intra-annual frequency. This research is valuable as it expands our understanding of a critically important and endangered biome in South America.

Response: Thank you for your thoughtful and positive feedback. We have carefully considered the comments raised in this review and have made revisions throughout our manuscript to address each point.

*General comments*

Overall, the manuscript is well-written, though certain sections could benefit from further clarification, and organizing content into subsections may help enhance readability. For example, the discussion is currently presented as a single section that addresses multiple aspects of ecosystem productivity: stems, canopy, fine root production, biomass allocation, and ecosystem-level NPP. The authors might consider restructuring these sections to better highlight their results. This suggestion also applies to the results section. Additionally, some passages could be refined to improve clarity, and certain sentences might benefit from adjustments in English style and grammar to enhance overall readability. Some of my general comments include:

Response: Thank you for your valuable and constructive feedback. We appreciate your suggestions on improving clarity and readability by incorporating subsections within the results and discussion sections. We agree that this restructuring, particularly within the discussion on ecosystem productivity, will better highlight key findings and enhance the manuscript's accessibility.

In line with your recommendation, we also refined certain passages and made adjustments in style and grammar in order to improve readability. We are confident that these revisions significantly strengthened the manuscript.

- **Introduction & main hypothesis:** What are the expected differences in how these plant communities respond to ENSO events, and what are the underlying reasons for these differences? The introduction currently lacks sufficient information to propose a well-developed hypothesis in this direction. It mainly describes basic characteristics of the plant communities without clearly linking

this information to the research objectives, making it difficult for the reader to grasp the relevance of these details. A critical missing element is a discussion on biomass allocation patterns in response to drought and their significance for productivity in the Cerrado, which is essential given that much of the discussion revolves around these patterns. Additionally, a more nuanced description of physiological differences beyond those briefly mentioned in lines 85-96 could better illustrate how these differences influence sensitivity to climate change, providing stronger context for the study's hypotheses. Addressing these gaps would require a comprehensive revision and restructuring of the introduction, including clearer research questions. Breaking down the introduction, results, and discussion into distinct sections could also improve the flow and clarity of the manuscript.

Response: We have restructured the introduction to provide a more comprehensive overview of biomass allocation patterns, particularly in relation to drought conditions and productivity in the Cerrado (lines 131-136). We recognise the importance of these patterns for understanding the research findings and aimed to highlight their significance more effectively. We have also expanded about physiological differences between plant communities, to better contextualise the anticipated sensitivities of these communities to climate change (lines 89-146).

We note, however, that there is currently very limited information available on the effects of ENSO specifically on the Cerrado, which restricts our ability to elaborate in this area fully. We have made an effort to incorporate as much relevant information as possible to strengthen the context of our hypotheses (lines 125-146).

- **NPP estimated metrics:** It is unclear why the authors estimate the contribution of herbivory to NPP and coarse root production, as these estimates are not used in any analysis or discussion. Including these estimates adds confusion and does not contribute significantly to the overall narrative, especially since these fluxes were not directly measured. Inferring their sensitivity to ENSO based on estimations with prescribed uncertainty could be highly misleading and does not add anything to the manuscript.

Response: We included these estimates to avoid underestimating NPP, following common practice in similar studies (as noted in GEM network publications, e.g. Malhi et al., 2009, 2015, 2017; Riutta et al., 2018). Including this component allows for a more accurate and comparable NPP estimate with other studies. Additionally, these estimates are scaled from other measured components and, as such, do not influence the interpretation of the interannual variation in NPP.

In light of your feedback, we have clarified this rationale in the manuscript to prevent any potential confusion (lines 245-251).

- **Cerrado 2018 productivity data:** The 2018 productivity data for the Cerrado is not clearly explained, and the authors only briefly touch upon the reasons for the

observed decline. Why was productivity so much lower in 2018? The authors suggest a lack of accuracy in the diameter readings (lines 367-370), raising concerns about the reliability of the data. This also prompts the question: was the data prior to 2018 accurate, or could similar measurement errors have led to overestimations and potentially flawed conclusions?

Response: We carefully reviewed the whole dataset and identified an error for 2018 in which measurements considered tree basal diameter instead of tree breast-height diameter. This was the explanation for that abrupt drop in productivity. We have accordingly corrected and recalculated all values, carefully double-checking the whole dataset.

We would like to emphasise that the data collected throughout the entire experiment, including prior to 2018, are considered reliable. We followed consistent quality control procedures and applied standardised protocols to all measurements, ensuring data accuracy across the study period.

- **Statistical analysis:** This section lacks critical information. The statistical models used for each analysis are not clearly described, leaving the reader uncertain about which model was applied to address specific research questions. This is particularly important because the data originate from only two plots (n=1 per forest type), raising questions about the sample size and analysis approach. Did the authors use subplots, litterfall traps, or individual trees as sampling units instead? If so, how was spatial autocorrelation within subplots accounted for? None of these methodological details are explained in the statistical analysis section, making it difficult to assess the robustness of the results.

Response: We appreciate the opportunity to clarify the statistical approach used in this study. We revised the manuscript to include these methodological details in the statistical analysis section, ensuring that the description is clear and transparent for readers (lines 332-345).

"Our analyses focused on comparing NPP across years (2014 to 2019), comprising the periods before, during, and after the El Niño 2015/2016 event, in both cerrado and cerradão. To compare total canopy NPP across years in each vegetation type (cerradão and cerrado), we performed a repeated-measures ANOVA. The statistical model considered year as a fixed factor, while litter traps were included as a random effect to account for the hierarchical structure of the data over time. When significant differences were detected, we used Tukey's post hoc test to compare total canopy NPP between years. We applied the same analysis to compare stem and fine root NPP across different years in each plot. For stem NPP, we used subplots as random effects, and for fine roots NPP, we used ingrowth cores as random effects. In cases where residuals violated ANOVA assumptions, we applied Friedman's non-parametric test."

- **Figures:** The figures need several enhancements to improve clarity. For example, pairwise comparisons are not indicated, making it difficult for readers to discern if values are significantly different. Some figures, like Figure 1, have font sizes that are too small, and others, like Figure 6, are missing error bars. In Figure 2, letters appear within the panels (upper left corner) that are not explained in the figure legend, leading to confusion. Although the manuscript specifies that the color scheme was chosen to be accessible to color-blind readers (lines ), many bar plots still use similar shades of green and include a dashed red line, which may not be effective. A simpler color scheme, such as using white and dark gray/black, might improve readability.

Response: Below are the actions we have included to address your suggestions:

Regarding pairwise comparisons, we included statistical differences in Figures 4, 5, and 6, with significant results clearly marked.

We have adjusted the font size in the figures, particularly in Figure 1, to ensure all text is legible.

We have removed letters in the panel from Figure 2 to improve clarity

Figure 6: We have excluded the tree mortality rate from the figure (and from the manuscript)

We have revised the colour scheme to ensure it is accessible to colour-blind readers. Although we initially chose a specific scheme for this purpose, we adjusted the bar colours to use higher contrast combinations.

*Line-by-line comments below:*

Lines 25-26: Perhaps you could also highlight whether these differences persisted during ENSO.

Response: We excluded this sentence to focus on the effect of El Niño on both plots.

Line 42: I think the neotropics in general experience the strongest drought on record. For instance tropical dry forest in Central America suffer exacerbated mortality rates, to the point that up 30% of all individuals of a given species died (Powers et al., 2020).

Response: We rephrased that sentence to incorporate this information (lines 47-48).

Line 66: This idea repeats one item (wildfires) in the list above. Maybe delete from the list and just state it here.

Response: Yes, it is above, but it is only listed as one of the problems in the region and here it is linked to El Niño (lines 68-72). So it is important to keep both.

Line 89: Perhaps you can use the word "indicates" instead of "means".

Response: We changed the paragraph to better support the hypotheses as suggested by reviewer 1 and that word was deleted (lines 88-116).

Lines 89-90: Is this a hypothesis you are proposing? One could argue that the earlier the leaf loss is due to more hydraulic vulnerability in these species (Reich & Borchert, 1984; Eamus, 1999; Brodribb et al., 2002; Sobrado, 2015; Vargas G. et al., 2021).

Response: To clarify, this is not a hypothesis but rather supporting information for our hypotheses, which are now explicitly stated. We have revised the text for clarity (lines 89–116).

Line 93: tree height and plant vulnerability to drought. Maybe you would like to check the work by Olson and collaborators (2018, 2020).

Response: We have included Olson et al. 2018 and also a relevant work on this topic in our study regions has been published and cited (Araujo et al., 2024) (lines 109-111).

Line 97: Maybe state the main hypothesis of the study here. Also, what are the broader implications of studying these responses. Can they inform how we predict the fate of this ecosystem in a warmer/drier world.

Response: We agree that stating the main hypotheses or expectations of the study at this point would strengthen the introduction. We revised the text and included our hypotheses, focusing on the effects of the El Niño event on productivity and allocation (lines 125-146).

We believe that discussing the broader implications, such as the fate of this ecosystem in a warmer or drier world, is more appropriate for the Discussion section. However, there are limitations to extrapolating short-term changes to long-term responses.

Line 108: What is the size of the plots, how many subplots are there? Something missing in the site description is the water table proximity, which is really important in this region of the world (Mattos et al., 2023).

Response: This information was in Table 1, but we have included it in the main text now (lines 230-298). Also, there is no evidence of a shallow water table (Marimon Junior & Haridasan, 2005).

Lines 157-159: Are you referring here to the hydrological year? Or is just the period being there was ENSO? I think using the hydrological year is the most appropriate (Aragão et al., 2007; Feng et al., 2013; Schwartz et al., 2020).

Response: We are using the hydrological year in this section, as defined by Aragão et al. (2007). We revised the text to clarify this point and ensure consistency with the reference to the hydrological year (lines 212-215).

Table 1: the writing in some of the descriptions is confusing (e.g., LAI).

Response: We are not using LAI information in this new version of the manuscript and have excluded this information.

Table 1: Loss to leaf herbivory and coarse root net primary productivity. Is the 1.37 the average of "shrubland" in Miranda et al. 2014? Since this was not measured and the systematic uncertainty seems arbitrary and do not follow the CV presented by Miranda et al. 2014, I would restrain of using this.

Response: The value of 1.37 is indeed based on the average for "shrubland" from Miranda et al. (2014), Table 4B. We acknowledge the limitations of applying this estimate and the associated uncertainty. However, these components are valuable for estimating total NPP, as they provide a more complete picture of the whole ecosystem productivity. While estimating total NPP is not the core aim of this paper, we find it useful to include this information, given the scarcity of such estimates for savannas. Additionally, a recent study by Terra et al. (2023) using 144 plots found a similar relationship, with a value of 1.58 for belowground productivity in Brazilian savannas. We have incorporated this reference to provide further context and support for the inclusion of these estimates in our discussion (line 275).

Lines 182-198: I would focus the estimations on the measurements you have data for. The problem of adding estimations with systematic errors, is that these can be highly inaccurate and can lead to misleading results. While I appreciate the intend to be as precise as possible in estimating the NPP. The focus of the study is to quantify the effects of ENSO on the measured fluxes. Adding all this information on unmeasured quantities is just confusing and does not seem to add much to the study.

Response: We only estimated leaf and coarse root herbivory. The information in lines 234-235 refers to parameters we did not measure. We included this information to provide context on the contribution of unmeasured components to total productivity.

Line 188: This is just the production of fine roots, because you are not considering the production of coarse roots.

Response: We agree with your observation and we changed this equation to: $NPP_{fine\ root} = NPP_{fine\ root}$ (line 307).

Lines 207-210: Maybe is worth explaining here the decision making process to define these % of error propagation.

Response: Here we were mostly following previous protocols and publications. For total NPP, the uncertainty value is calculated by combining uncertainty of each component, by error propagation (Hughes & Hase, 2010; Malhi et al., 2015). So NPP_total_se is a mixture of systematic and random errors. For each component, we have provided more texts to explain the logic. For example, for NPP litter fall, we use standard error, instead of an arbitrary systematic error because we believe that systematic error is trivial for NPP litter fall (and a lot more explanations, please check them in the method section)

Lines 215-216: Please describe the models more. Main effects? nested effects? Interactions? Also, how did you deal with a sample size of 1 per forest type?

Response: To clarify, we changed the text to (lines 332-345): "Our analyses focused on comparing NPP across years (2014 to 2019), comprising the periods before, during, and after the El Niño 2015/2016 event, in both *cerrado* and *cerradão*. To compare total canopy NPP across years in each vegetation type (*cerradão* and *cerrado*), we performed a repeated-measures ANOVA. The statistical model considered year as a fixed factor, while litter traps were included as a random effect to account for the hierarchical structure of the data over time. When significant differences were detected, we used

Tukey's post hoc test to compare total canopy NPP between years. We applied the same analysis to compare stem and fine root NPP across different years in each plot. For stem NPP, we used subplots as random effects, and for fine roots NPP, we used ingrowth cores as random effects. In cases where residuals violated ANOVA assumptions, we applied Friedman's non-parametric test. All analyses were conducted in the R environment, with a significance level of 0.05.".

Lines 220-221: This is unnecessary, and shades of green with red dashed lines is not a color-blind friendly combination of colors.

Response: We agree with your observation and excluded this information from the manuscript.

Line 229: Can we say then that ENSO did not affect the Cerrado?

Response: Yes, we can say that ENSO did not affect the overall productivity of the cerrado.

Line 240: What are these little letters in the upper left corners, what do they represent?

Response: These elements were unnecessary and we excluded.

Lines 250-253: This is not correct. First, 3 b) shows dbh growth, not NPP of alive trees. Second, npp should consider the biomass lost to mortality (Anderson-Teixeira et al., 2016).

Response: Good point. We have decided to delete DBH growth (this Figure 3B) and focus only on productivity.

Line 260: Can you indicate whether you observed differences in the plot?

Response: Done.

Line 276: Again, indicate with letters pairwise comparisons.

Response: Done.

Line 291: Indicate significance here.

Response: We included letters to indicate significance among years to total components.

Lines 310-312: This is not clear from the figures.

Response: We changed the text and also included more information (lines 377-384). We have also included the allocation text along with the total productivity (Figure 2; section "3.1 Total NPP and its allocation") as suggested by reviewer 2 and we believe it is now easier to understand.

Line 318: Add the color scale to the plot as a legend, the axes should be black and same with the gridlines.

Response: We implemented the necessary changes.

Line 325: This is not clear from the figures. Resistant, yes when comparing absolute values. Did you try looking at the relative change in productivity? Resilient, is impossible to compare when the 2018 data seem wrong.

Response: We appreciate your suggestion to consider relative changes in productivity and have incorporated additional information in the text (lines 393-394). Our analysis of relative changes confirms that stem NPP was affected in the cerradão but remained stable in the cerrado (lines 421-432). We revised the text to clarify this finding and enhance the interpretation of the results (lines 482-494).

Additionally, we have verified the 2018 data by rechecking the diameter measurements and have now included the correct values, as explained above.

Lines 335-338: This is unnecessary, and just fill space.

Response: We agree and exclude (lines 497-503).

Lines 343-344: dominant species or most important species might be the most appropriate term.

Response: We changed to "dominant".

Line 345: plants can adjust this... See Guo et al (2020) (https://doi.org/10.1111/nph.16196) and Guo et al. (2024) (https://doi.org/10.1111/nph.19805). Perhaps is not a matter of adjusting their stomatal control, but more about their hydraulic safety margin? Nowhere in the discussion or introduction hydraulic safety margins are mentioned. Also, in Jankoski et al. 2022, Tachigali vulgaris seems to be partially isohydric (Fig. 3), which suggests some degree of stomatal regulation.

Response: We excluded the results and discussion on tree mortality to focus on productivity.

Lines 349-351: How does the amount of embolisms link to wind resistance? In your previous article (Reis et al. 2022), you assessed mortality but did not gather enough information to suggest that drough+heat decreased the capacity of stems to resist winds. Please explain?

Response: We excluded the results and discussion on tree mortality to focus on productivity.

Lines 354-355: higher productivity when?

Response: We excluded the results and discussion on tree DBH growth to focus on productivity.

Lines 362-364: This is not clear from that reference....

Response: Again, we excluded the results and discussion on tree DBH growth and mortality to focus on productivity.

Lines 367-370: plants were competing also before ENSO. This statement does not make sense. Was there a fire event after or during ENSO???? The one problem is the magnitude of the inaccuracy, that is concerning.

Response: We excluded the results and discussion on tree DBH growth to focus on productivity. Regarding fire events, the area did not experience any fires during or after the El Niño event. The last recorded fire in the area occurred in 2008.

Lines 377-378: This could also be root phenological patterns associated to water availability (Kummerow et al., 1990; Kavanagh & Kellman, 1992).

Response: We included the text (lines 655-657): "The strategy observed in the cerrado was similar to that of tropical dry forests, reflecting root phenological patterns linked to water availability (Kummerow et al., 1990; Kavanagh & Kellman, 1992)."

Lines 388-390: Yes, but this was not measured. Something the authors have not developed is whether plants are deciduous or evergreen or brevideciduous and how the underlying physiological differences link to observed responses. There are only a couple of lines about that, but that might have implications in the different responses of the two plant communities. For example:

- It might be important in the regulation of water transport and photosynthesis (Brodribb et al., 2002).

- It might also be related with rooting depth (Smith-Martin et al., 2019, 2020).

Response: We acknowledge that we did not collect data on the deciduousness of the species, which limits our ability to fully explore the role of leaf phenology in the observed responses. Our intention in discussing leaf loss during the El Niño year was to highlight its potential contribution to reduced productivity, as there is a well-established relationship between leaf area and plant growth (Zhang et al., 2014). We have clarified this in the revised manuscript (lines 532-567).

Line 396: this terminology (wood-fine root trade-offs) is confusing as is not defined. The manuscript will benefit from better staging of these concepts in the introduction.

Response: Sorry for the confusion. We have removed the term 'trade-off', which is not necessary because readers would understand fine with allocation to wood, root, leaves etc. Now the introduction mentions allocation and hypotheses about allocation without saying 'trade-off'.

Lines 397-400: This is an interesting proposition, but it will benefit from more background information. Basically, can we expect different biomass allocation patterns among biomes?

Response: Given that there is a huge spatial dynamics in NPP allocation within tropical ecosystems (Zhang-Zheng et al., 2024), with a space for time assumption, one would hypothesise NPP allocation change under El Nino. We have provided more context in this paragraph (lines 512-517).

Line 407: Maybe "important" is more appropriate than "vital".

Response: Done.

Line 411: Are you referring to ENSO or drought?

Response: We are referring to El Niño (ENSO). The text has been revised to clarify this (line 674).

Line 413: this was not quantified.

Response: We have revised the sentence to include the phrase "for other forest types in a recent study in the region (Gatti et al., 2021)". Lines 675-676.

Lines 414-415: Why is this? It just appear out of the bleu.

Response: We have revised the text to clarify the role of the *cerradão* as a transitional vegetation type within the Amazon-Cerrado ecotone (ex. lines 25, 83, 150).

*References*

Anderson-Teixeira KJ, Wang MMH, McGarvey JC, LeBauer DS. 2016. Carbon dynamics of mature and regrowth tropical forests derived from a pantropical database (TropForC-db). Global Change Biology 22: 1690–1709.

Aragão LEOC, Malhi Y, Roman-Cuesta RM, Saatchi S, Anderson LO, Shimabukuro YE. 2007. Spatial patterns and fire response of recent Amazonian droughts. Geophysical Research Letters 34.

Brodribb TJ, Holbrook NM, Gutiérrez MV. 2002. Hydraulic and photosynthetic co-ordination in seasonally dry tropical forest trees. Plant, Cell & Environment 25: 1435–1444.

Eamus D. 1999. Ecophysiological traits of deciduous and evergreen woody species in the seasonally dry tropics. Trends in Ecology & Evolution 14: 11–16.

Feng X, Porporato A, Rodriguez-Iturbe I. 2013. Changes in rainfall seasonality in the tropics. Nature Climate Change 3: 811–815.

Kavanagh T, Kellman M. 1992. Seasonal Pattern of Fine Root Proliferation in a Tropical Dry Forest. Biotropica 24: 157.

Kummerow J, Castillanos J, Maas M, Larigauderie A. 1990. Production of fine roots and the seasonality of their growth in a Mexican deciduous dry forest. Vegetatio 90: 73–80.

Mattos CRC, Mazzochini GG, Rius BF, Penha D, Giacomin LL, Flores BM, Silva MC, Xavier RO, Nehemy MF, Petroni AR, et al. 2023. Rainfall and topographic position determine tree embolism resistance in Amazônia and Cerrado sites. Environmental Research Letters 18: 114009.

Olson ME, Anfodillo T, Rosell JA, Martínez-Méndez N. 2020. Across climates and species, higher vapour pressure deficit is associated with wider vessels for plants of the same height. Plant, Cell & Environment 43: 3068–3080.

Olson ME, Soriano D, Rosell JA, Anfodillo T, Donoghue MJ, Edwards EJ, León-Gómez C, Dawson T, Camarero Martínez JJ, Castorena M, et al. 2018. Plant height and hydraulic vulnerability to drought and cold. Proceedings of the National Academy of Sciences 115: 7551–7556.

Powers JS, Vargas G. G, Brodribb TJ, Schwartz NB, Pérez-Aviles D, Smith-Martin CM, Becknell JM, Aureli F, Blanco R, Calderón-Morales E, et al. 2020. A catastrophic tropical drought kills hydraulically vulnerable tree species. Global Change Biology 26: 3122–3133.

Reich PB, Borchert R. 1984. Water Stress and Tree Phenology in a Tropical Dry Forest in the Lowlands of Costa Rica. Journal of Ecology 72: 61–74.

Schwartz NB, Lintner BR, Feng X, Powers JS. 2020. Beyond MAP: A guide to dimensions of rainfall variability for tropical ecology. Biotropica 52: 1319–1332.

Smith-Martin CM, Bastos CL, Lopez OR, Powers JS, Schnitzer SA. 2019. Effects of dry-season irrigation on leaf physiology and biomass allocation in tropical lianas and trees. Ecology 100: e02827.

Smith-Martin CM, Xu X, Medvigy D, Schnitzer SA, Powers JS. 2020. Allometric scaling laws linking biomass and rooting depth vary across ontogeny and functional groups in tropical dry forest lianas and trees. New Phytologist 226: 714–726.

Sobrado MA. 2015. Leaf Tissue Water Relations Are Associated with Drought-Induced Leaf Shedding in Tropical Montane Habitats. American Journal of Plant Sciences 6: 2128–2135.

Vargas G. G, Brodribb TJ, Dupuy JM, González-M. R, Hulshof CM, Medvigy D, Allerton TAP, Pizano C, Salgado-Negret B, Schwartz NB, et al. 2021. Beyond leaf habit: generalities in plant function across 97 tropical dry forest tree species. New Phytologist 232: 148–161.

References:

Aragão, L. E. O., Malhi, Y., Roman-Cuesta, R. M., Saatchi, S., Anderson, L. O., & Shimabukuro, Y. E. (2007). Spatial patterns and fire response of recent Amazonian droughts. Geophysical Research Letters, 34(7).

Araújo, I., Marimon, B. S., Junior, B. H. M., Oliveira, C. H., Silva, J. W., Beú, R. G., ... & Galbraith, D. R. (2024). Taller trees exhibit greater hydraulic vulnerability in southern Amazonian forests. Environmental and Experimental Botany, 226, 105905.

Hughes, I., & Hase, T. (2010). Measurements and their uncertainties: a practical guide to modern error analysis. OUP Oxford.

Malhi, Y., Aragao, L. E. O., Metcalfe, D. B., Paiva, R., Quesada, C. A., Almeida, S., ... & Teixeira, L. M. (2009). Comprehensive assessment of carbon productivity, allocation and storage in three Amazonian forests. Global Change Biology, 15(5), 1255-1274.

Malhi, Y., Doughty, C. E., Goldsmith, G. R., Metcalfe, D. B., Girardin, C. A., Marthews, T. R., ... & Phillips, O. L. (2015). The linkages between photosynthesis, productivity, growth and biomass in lowland Amazonian forests. Global Change Biology, 21(6), 2283-2295.

Malhi, Y., Girardin, C. A., Goldsmith, G. R., Doughty, C. E., Salinas, N., Metcalfe, D. B., ... & Silman, M. (2017). The variation of productivity and its allocation along a tropical elevation gradient: a whole carbon budget perspective. New Phytologist, 214(3), 1019-1032.

Marimon Junior, B. H., & Haridasan, M. (2005). Comparação da vegetação arbórea e características edáficas de um cerradão e um cerrado sensu stricto em áreas adjacentes sobre solo distrófico no leste de Mato Grosso, Brasil. Acta Botanica Brasilica, 19, 913-926.

Riutta, T., Malhi, Y., Kho, L. K., Marthews, T. R., Huaraca Huasco, W., Khoo, M., ... & Ewers, R. M. (2018). Logging disturbance shifts net primary productivity and its allocation in Bornean tropical forests. Global Change Biology, 24(7), 2913-2928.

Terra, M. C., Nunes, M. H., Souza, C. R., Ferreira, G. W., do Prado-Junior, J. A., Rezende, V. L., ... & de Mello, J. M. (2023). The inverted forest: Aboveground and notably large belowground carbon stocks and their drivers in Brazilian savannas. Science of The Total Environment, 867, 161320.

Zhang-Zheng, H., Adu-Bredu, S., Duah-Gyamfi, A., Moore, S., Addo-Danso, S. D., Amissah, L., ... & Malhi, Y. (2024). Contrasting carbon cycle along tropical forest aridity gradients in West Africa and Amazonia. Nature Communications, 15(1), 3158.

##################################################################

Referee #2

In this manuscript, the authors examined how the vegetation in the Cerrado responded to the 2015-16 El Niño event, using field measurements collected between 2014 and 2019. They discovered contrasting patterns between the cerrado and cerradão ecosystems. The dataset employed in this study is particularly valuable and unique, with intriguing results. Nevertheless, there are two major issues that the authors need to address during the revision.

Response: Thank you for your positive feedback. We have carefully considered the comments raised in this review and made revisions throughout our manuscript to address each point.

1. The 'Results' and 'Discussion' sections must be better structured. At present, the authors present all the information without using sub-headings, particularly in the 'Discussion' section.

Below are the suggestions for the structure of the 'Results' section.

3.1 Total NPP and its allocation

3.2 Canopy NPP

3.3 Stem NPP and mortality

3.4 Root NPP

3.5 Dynamics among canopy, stem and root NPP

The 'Discussion' should adopt a similar structure.

Response: We have revised the Results and Discussion sections to organise them into thematic topics, as well as restructured the Materials and Methods section. We hope these changes enhance the clarity and readability of the manuscript.

2. The authors defined the 12 months from May 2015 to April 2016 as the 2015-16 El Niño event (lines 157-158) and presented temperature, precipitation, and MCWD anomalies during this period compared to other years (Fig. 1). Why not use this same May-to-April definition for the remaining figures in the study? This would provide a clearer understanding of the 2015-16 El Niño event's impact on vegetation.

Response: Some variables, such as fine root and branch NPP, are measured quarterly, while others, such as stem NPP and stem mortality, are measured annually. This makes it challenging to calculate NPP for the period from May to April. On the other hand, variables like litterfall are measured monthly, enabling such calculations. We have analysed the data both on an annual basis and using a May-to-April calendar year. However, the results were similar because plants typically take time to respond to climatic phenomena.

We chose to present the data on an annual basis because adopting a May-to-April timeframe would result in the loss of one year of data: four months from the first year (before May) and eight months from the final year (after April). Since the results are comparable, losing an entire year of data collection would not be advantageous.